# Serum anti-nucleocapsid antibody correlates of protection from SARS-CoV-2 re-infection regardless of symptoms or immune history
Sho Miyamoto [1,2], Koki Numakura[1], Ryo Kinoshita[3], Takeshi Arashiro[1,2], Hiromizu Takahashi[4], Hiromi Hibino[5], Minako Hayakawa[1], Takayuki Kanno [1], Akiko Sataka[1], Rena Sakamoto[1], Akira Ainai[1], Satoru Arai[6], Motoi Suzuki[7], Daisuke Yoneoka[3], Takaji Wakita[8] & Tadaki Suzuki [1,2] ✉

## Abstract

**Background** High spike-based vaccine coverage led to a high seroprevalence of anti-spike (S) antibodies against SARS-CoV-2 in Japanese adults in 2024. Nevertheless, the COVID-19 epidemic continues, and individuals with hybrid immunity are becoming more common in these populations.

**Methods** We conducted a prospective cohort study to measure serum anti-SARS-CoV-2 antibody levels in 4496 Japanese adults as part of the national seroepidemiological survey. This study evaluated the correlation between first-visit anti-SARS-CoV-2 antibody levels and their effectiveness in providing protection until the second visit during the Omicron BA.5 epidemic.

**Results** Reduced symptomatic infection risk was found to be associated with anti-S antibody, anti-nucleocapsid (N) antibody, and BA.5 neutralizing antibody levels. However, the reduced asymptomatic infection risk associated with anti-S antibody or BA.5 neutralizing antibody levels was limited. In contrast, higher anti-N antibody levels were strongly linked to a reduced asymptomatic infection risk. Furthermore, higher anti-N antibody levels were also associated with a reduced risk of re-infection in individuals with hybrid immunity.

**Conclusion** These observations highlight the potential of anti-N antibody level as a correlate of protection against SARS-CoV-2 asymptomatic infection and re-infection. The findings indicate that individuals with hybrid immunity have a distinct protective immunity against both symptomatic and asymptomatic infection beyond serum anti-S and neutralizing antibodies against circulating viral strains, which correlate with serum anti-N antibodies.

## Plain language summary

During the COVID-19 pandemic, vaccines helped many people develop antibodies, proteins that fight infections, against the virus. However, the virus continues to spread, and some people have both vaccine-induced and infection-induced immunity, known as hybrid immunity. This study used 4496 Japanese adults to understand how different antibodies protect against COVID-19, especially during the Omicron BA.5 wave. They measured levels of two antibody types (anti-S and anti-N) in the participants' blood. The study found that higher levels of anti-N antibodies were linked to a lower risk of getting infected again, even without symptoms. This suggests that anti-N antibodies are important for protection, especially in people with hybrid immunity. This suggests that future COVID-19 vaccines could be improved by focusing on inducing strong anti-N antibody responses or mimicking hybrid immunity, which might offer better protection against re-infections.

Several spike-based vaccines against SARS-CoV-2 with mRNA or viral vector modalities were developed during the early COVID-19 pandemic and showed high efficacy in early clinical trials and during the pre-Omicron epidemic period[1–4]. However, since the emergence of Omicron variants at the end of 2021, Omicron sublineages with high resistance to humoral immunity induced by spike-based vaccines have been reported[5,6], leading to a decline in vaccine effectiveness against infection[7,8]. During the pre-Omicron epidemic period, anti-spike (S) antibody titers against the ancestral strain induced by vaccination were identified as immunological correlates for protection against SARS-CoV-2 infection and severe disease[4,9,10]. While the prevention of severe disease through vaccination was confirmed even during the Omicron BA.1/2 and BA.4/5 epidemic periods[11], a higher anti-spike (S) antibody titer against the ancestral strain was required for protection against infection during these periods compared to the pre-Omicron epidemic period[12–16]. Omicron sublineages tend to be selected for mutations with high humoral immune evasion capabilities[17],

and the protective effect of anti-S antibody titers against the ancestral strain needs to be re-evaluated in response to changes in the antigenicity of the emerging variants[18]. Recently, with the increase in the proportion of infected individuals, it has been reported that not only the neutralizing antibody titers and anti-S antibody titers induced by vaccination but also past infection history are significantly associated with a reduction in infection risk[19,20]. Recent reports have shown that nasal mucosal secretory anti-spike IgA antibody levels, elicited after infection, are associated with prevention of the SARS-CoV-2 infection and infectious viral shedding in the upper respiratory tract[21,22]. Therefore, to accurately estimate the extent of the COVID-19 epidemic in the post-COVID-19 pandemic era, it is necessary to determine the immunological correlates of protection against SARS-CoV-2 re-infection in individuals with hybrid immunity due to both vaccination and infection[23].

Evaluating the accurate potential of immunological correlates for preventing infection requires the assessment of asymptomatic and undiagnosed re-infected cases, which also influence the COVID-19 epidemic dynamics. Recently, undiagnosed re-infections were reported to be prevalent, implying that analyses focusing only on symptomatic infections risk underestimating the re-infection risk[24,25]. Against this backdrop, we conducted this prospective cohort study as part of the national COVID-19 seroepidemiological survey in Japan that involved two blood samplings and antibody tests during the Omicron BA.5 epidemic period (https://cov-spectrum.org) from December 2022 to February 2023. By identifying infected individuals through diagnosis and seroconversion of infection-derived anti-nucleocapsid (N) antibodies, we included undiagnosed primary infections and re-infections as newly infected cases during the study period. We evaluated the infection risk reduction associated with anti-S antibody titers and BA.5 neutralizing antibody titers and the combined effect of infection-related anti-N antibody levels during the Omicron BA.5 epidemic period by analyzing the association of combined serum antibody levels with the occurrence of a new infection. This study shows that serum anti-N antibodies are robust immune factors that correlate with protection against SARS-CoV-2 re-infection regardless of symptoms, even in individuals with hybrid immunity. These results indicate that the immunity induced by vaccination and infection strongly elicits protective immune factors against infections, including symptomatic and asymptomatic infections.

## Methods

### Survey design, participants, and ad hoc study design

For the national COVID-19 seroepidemiological survey, residents of Miyagi, Tokyo, Aichi, Osaka, and Fukuoka prefectures were randomly selected using the Basic Resident Register via multistage sampling, as described previously[26]. Sampling and data collection were conducted over several days at 15 sites (three municipalities in each of the five prefectures). For each prefecture, at least one municipality from each of the following three municipality types was chosen for the surveys: small (<100,000 population), medium (≥100,000 population), and large (ordinance-designated city/special ward). The samples were divided according to the relative population sizes of the municipalities, and residents were randomly sampled from each municipality. We planned to enroll 15,000 individuals from five prefectures (3000 individuals per prefecture). Assuming a response rate of 20%, we randomly sampled 75,000 individuals aged 20 years or older and invited them via mail. Only one participant from each household participated in the study. No financial incentives are provided for the participants except for feedback on the serologic test results. Individuals who agreed to participate in the study answered a self-administered questionnaire, visited the designated site where they provided written consent, and had their blood drawn. Briefly, the questionnaire included demographic information (age, biological sex, occupation, municipality, etc.), comorbidities, COVID-19 vaccination status, and history of SARS-CoV-2 infection. The survey was conducted as a public health investigation under the Act on the Prevention of Infectious Diseases and Medical Care for Patients with Infectious Diseases (Infectious Diseases Control Law) and planned by the Ministry of Health,

Labor, and Welfare of Japan (MHLW) and the National Institute of Infectious Disease (NIID). The survey assessed the prevalence of anti-N and anti-S antibodies. The MHLW randomly selected the potential study participants and mailed invitations to them to participate in the survey, which was carried out as a public health investigation under the Infectious Diseases Control Law. Descriptive results of the national COVID-19 seroepidemiological survey for the entire cohort have been published in Japanese on the MHLW/NIID websites[27].

The ad hoc study, as a research activity, evaluated the protective efficacy of anti-N antibodies against re-infection, based on the survey data. Consent for this ad hoc study was obtained as part of the survey consent process, which included agreement to use data for further research.

### Ethical approval

All the samples, protocols, and procedures described herein were approved by the Medical Research Ethics Committee of the NIID and involved human participants, based on the principles of the Declaration of Helsinki (approval numbers 1457, 1472, and 1730).

### Definition of newly infected individuals

The study included 5627 individuals who participated in two consecutive tests. To fairly examine whether antibody levels at the beginning of the observation period (baseline antibody levels) were associated with a reduced risk of infection during the two-month observation period, those who were vaccinated during the observation period were excluded (1044 individuals). To use anti-N antibody seroconversion to determine infection, we excluded 87 individuals with a history of infection within 30 days of the initial antibody test (87 individuals). Consequently, there were 4496 subjects eligible for the study.

Participants with a history of self-test confirmed SARS-CoV-2 infection during the observation period, those who were diagnosed with COVID-19 during the observation period, or those with a seroconverted anti-N antibody from the initial to the second test (positive threshold, 1.0 COI) were considered newly infected cases. Participants who showed a four-fold or higher increase in anti-N antibody levels at the second antibody test compared to those at the initial test were considered newly infected, including those with re-infections (Fig. 1b, d). Specifically, participants with no previous COVID-19 diagnosis, or previous self-test confirmed SARS-CoV-2 infection in the self-administered questionnaire at the time of the December 2022 survey (from November 26, 2022, to December 27, 2022) and whose serum anti-N antibody level at the December 2022 survey was below 1.0 COI, were considered uninfected participants at the beginning of the observation period. Among the uninfected participants at the beginning of the observation period, those who subsequently reported a COVID-19 diagnosis or self-test confirmed SARS-CoV-2 infection at the February 2023 survey (from February 3, 2023, to March 4, 2023) or whose serum anti-N antibody level at the February 2023 survey was above 1.0 COI, were defined as newly infected cases during the observation period from December 2022 to February 2023. Among the prior-infected participants at the beginning of the observation period, those who reported a COVID-19 diagnosis, or self-test confirmed SARS-CoV-2 infection between the initial and the second survey, were considered newly re-infected during the observation period. Based on the anti-N antibody fold increase model for re-infections, participants whose serum anti-N antibody levels increased by four-fold or more from the December 2022 survey to the February 2023 survey were considered as newly re-infected.

### Definition of symptomatic and asymptomatic infections

Symptomatic individuals with COVID-19 were defined as individuals with any of the following symptoms, based on a previous study[8]: malaise, chills, joint pain, headache, runny nose, cough, sore throat, shortness of breath, gastrointestinal symptoms (vomiting, diarrhea, or stomach ache), and loss of taste or smell.

Participants who met the criteria for newly infected individuals described above, along with COVID-19 symptoms described in the

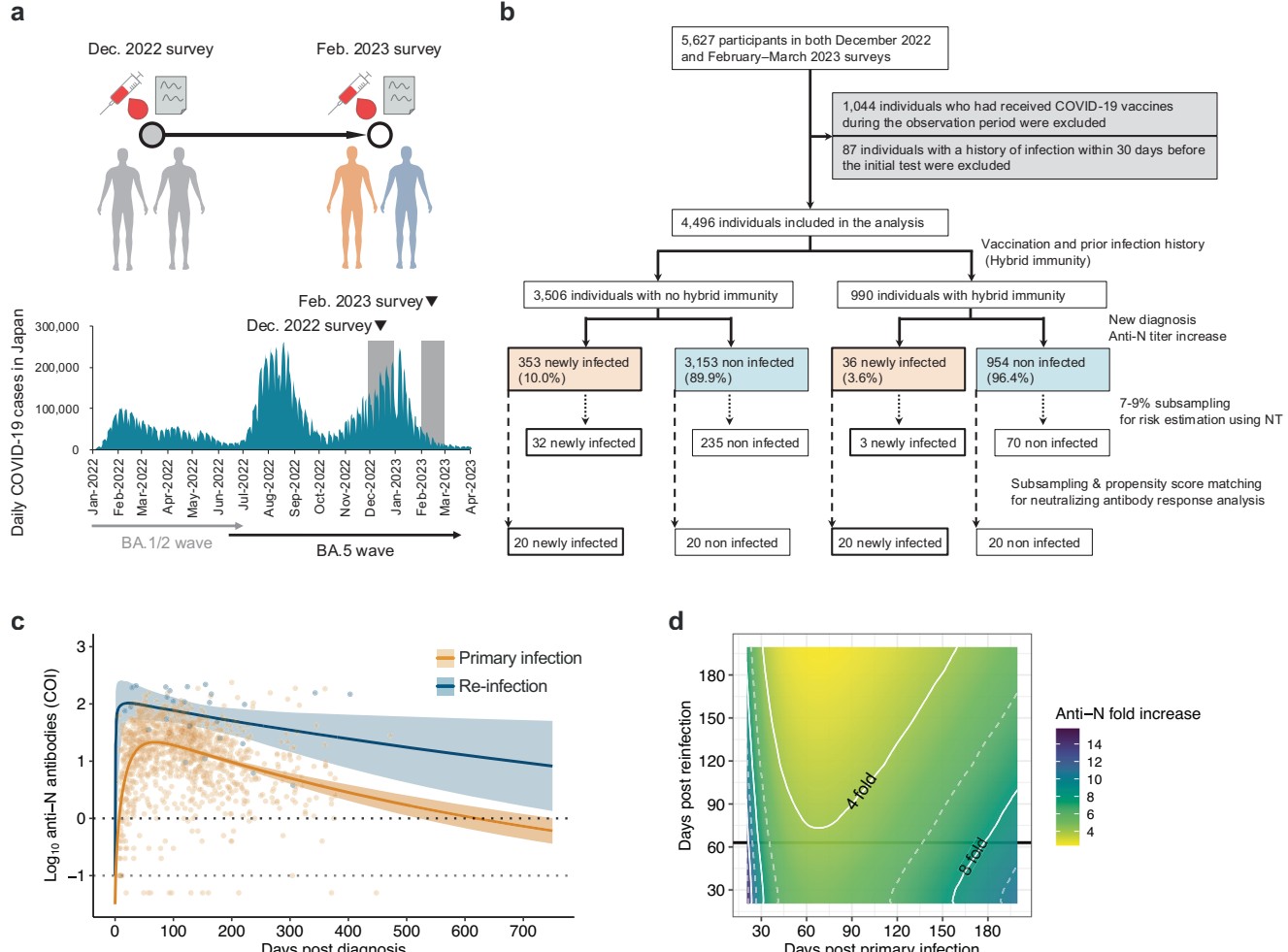

**Fig. 1 | Schematic diagram of this study and serum anti-N antibody level dynamics. a** Overview of the seroepidemiological survey conducted in Japan for this study. Serum samples and self-administered questionnaires were collected in two rounds conducted in December 2022 and February 2023. Sampling and data collection were carried out over several days at 15 sites (three municipalities in each of five prefectures). The periods of the two surveys correspond to the gray areas in the graph of COVID-19 case counts in Japan (https://www.mhlw.go.jp/stf/covid-19/open-data.html). The December 2022 survey was conducted from November 26, 2022, to December 27, 2022, while the February 2023 survey was conducted from February 3, 2023, to March 4, 2023. **b** Flow diagram of the study design and inclusion of participants. See Method for details. **c** A model of the dynamics of anti-N antibody

levels in diagnosed primary infection ($n = 1218$) and re-infection ($n = 26$). Each data point (dots), along with the median (lines) and 95% credible intervals (CredIs) (ribbon) for the mean anti-N antibody levels, is shown. The dark gray dotted line indicates the cutoff index (1.0 COI) according to the manufacturer's manual, and the light gray dotted line represents the detection limit (0.1 COI). **d** Estimation of the fold increase in anti-N antibody levels induced by re-infection. The fold increase was calculated from the ratio of estimated anti-N antibody levels on the days post-primary diagnosis (infection) at re-infection to the titers at corresponding days post-re-infection. The white dotted and solid lines show a 2× and 4× fold increase, respectively, in the anti-N antibody level. The black line shows the median observation period (63 days).

February 2023 survey questionnaires, were classified as newly symptomatic infected individuals. Participants with a history of self-test confirmed SARS-CoV-2 infection or COVID-19 diagnosis during the observation period without COVID-19 symptoms described in the February 2023 survey questionnaires were classified as newly "asymptomatic SARS-CoV-2 test confirmed" infected individuals, as shown in Supplementary Table 2. Among asymptomatic infected individuals, participants without a history of COVID-19 diagnosis or self-test confirmed SARS-CoV-2 infection during the observation period but met the criteria for newly infected individuals through anti-N antibody level increases (seroconverted anti-N antibody (positive threshold, 1.0 COI) or a four-fold increase from the December 2022 survey to the February 2023 survey) were classified as newly "asymptomatic serological defined" infected individuals, as shown in Supplementary Table 2. For infection risk estimation (Figs. 3–5, Supplementary Fig. 3, Supplementary Table 3), the asymptomatic infection category included both "asymptomatic SARS-CoV-2 test confirmed" and "asymptomatic serologically defined" infected individuals.

## Subsampling for neutralizing antibody titration
To evaluate antibody responses to the SARS-CoV-2 ancestral strain and Omicron BA.5 variant before and after infection, we randomly selected 20 newly infected individuals with hybrid immunity and 20 without hybrid immunity. Furthermore, we performed nearest neighbor propensity score matching for these 20 individuals using binary variables for age groups (20–64 years or 65 years or above), biological sex, presence of comorbidities, and Omicron BA.1/BA.5 bivalent vaccination to subsample 20 non-infected individuals from the observation period for each group (Fig. 1b and Supplementary Table 4). Propensity score matching for the subsampling was performed using *MatchIt* 4.5.5 with the default setting.

To estimate the risk of infection using neutralizing antibody titers against BA.5, we performed a stratified random sampling of 7–9% of newly infected and non-infected individuals during the observation period, both for those with and without hybrid immunity (Fig. 1b and Supplementary Table 5).

### Electrochemiluminescence immunoassay

Serum samples were heat-inactivated at 56 °C for 30 min before use. Antibody titers for the ancestral spike (S) receptor-binding domain and nucleocapsid (N) were measured using Elecsys Anti-SARS-CoV-2 S (Roche, Basel, Switzerland) and Elecsys Anti-SARS-CoV-2 (Roche) kits, respectively, according to the manufacturer's instructions. A cutoff index (COI) of 1.0 and a cutoff value of 0.8 BAU/ml, as determined by the manufacturer, were used to determine the presence/absence of anti-N antibody levels and anti-S antibody titers, respectively. Since the COVID-19 vaccines approved in Japan are only spike-based vaccines, anti-N antibodies are induced by infection but not by vaccines, whereas anti-S antibodies are induced by both infection and vaccines. The numerical results in U/mL of the Elecsys Anti-SARS-CoV-2 S assay and the WHO BAU/mL are equivalent.

### Cells

VeroE6/TMPRSS2 cells (JCRB1819, Japanese Collection of Research Bioresources Cell Bank; Osaka, Japan) were maintained in low-glucose Dulbecco's modified Eagle's medium (DMEM) (Fujifilm, Osaka, Japan) supplemented with 10% heat-inactivated fetal bovine serum (FBS) (Biowest, Nuaillé, France), 1 mg/mL geneticin (Thermo Fisher Scientific), and 100 units/mL penicillin/streptomycin (Thermo Fisher Scientific) at 37 °C in an atmosphere of 5% $CO_2$.

### Live virus neutralization assay

The SARS-CoV-2 ancestral strain, WK-521 (lineage A, GISAID ID: EPI_ISL_408667), and the Omicron variant, TY41-702 (lineage BA.5/BE.1, GISAID: EPI_ISL_13241867), were used. Live virus neutralization assays were performed as described previously[6]. In brief, serum samples were serially diluted (via two-fold dilutions starting from 1:5) in high-glucose DMEM supplemented with 2% FBS and 100 units/mL penicillin/streptomycin and were mixed with 100 $TCID_{50}$ SARS-CoV-2 viruses, followed by incubation at 37 °C for 1 h. The virus-serum mixtures were added to the VeroE6/TMPRSS2 cells seeded in 96-well plates and cultured at 37 °C in the presence of 5% $CO_2$ for five days. After culturing, the cells were fixed using 20% formalin (Fujifilm) and stained with crystal violet solution (Sigma–Aldrich, St. Louis, MO, USA). Neutralization titers (NT) were defined as the geometric mean of the reciprocal of the highest sample dilution that protected at least 50% of the cells from a cytopathic effect in duplicate series. As the sera from individuals were limited in quantity, this assay was performed only once. All experiments using live viruses were performed in a biosafety level 3 laboratory at the NIID.

### Infection risk estimation

The immune correlates of the infection risk analysis were conducted based on a previous study[10]. Log-transformed antibody titers were analyzed using Bayesian generalized additive models (GAM) for binary data, with cubic spline smoothing applied to antibody titers to allow a nonlinear effect. No adjustments were made for multiple comparisons. Combined models were fitted for combinations of anti-N antibody levels with anti-S antibody titers (Figs. 3a–i, 5a–c, Supplementary Fig. 3) or BA.5 NT (Figs. 4, 5d–f), controlling for baseline exposure risk, and weighted using inverse probability weights as described below. The definition of new infection, regardless of symptoms, newly symptomatic infection, and newly asymptomatic infection, was described above.

In addition to antibody titers, we assumed that each participant's absolute risk of new infections depends on the region and basic demographics. We used a Poisson regression model to predict the probability of infection, adjusting for the participants' baseline characteristics, and reducing potential bias in the availability of samples between newly infected and non-infected individuals. The predictors were age group (20–64 years or 65 years or older), biological sex, presence of comorbidities, Omicron BA.1/BA.5 bivalent vaccination, vaccination count (categorized as 0 to 5 doses), prior infection history, and municipality. We used the estimated probability as these inverse probability weights in the above GAM to adjust for the potential bias from participants' baseline characteristics to better isolate the

association of immune correlates (antibody titers) on infection risk reduction.

The infection risk estimation in the GAM was constructed using *brms* 2.21. In the GAM, a Poisson distribution was used for the response variable. The newly infected response variable was modeled as a function of smooth terms for combinations of anti-N antibody levels with anti-S antibody titers or BA.5 NT using cubic splines with three basis functions (k = 3). Parameter estimation was performed using the Markov chain Monte Carlo (MCMC) approach implemented in *rstan* 2.26 (https://mc-stan.org). Four independent MCMC chains were run with 5000 steps, including a warm-up period of 1000 steps, with subsampling every five iterations. We confirmed that all the estimated parameters showed <1.01 R-hat convergence diagnostic values and >1600 effective sampling size values, indicating that the MCMC runs were convergent. Information on the model estimates is summarized in Supplementary Tables 6–10. Due to the small number of symptomatic cases among individuals with hybrid immunity, stable estimation was not achieved in the newly symptomatic infection risk estimation.

In this cohort, the overall risk of a control group with no vaccination or prior infection history included in the individuals with non-hybrid immunity, referred to as no exposures, was calculated using 16 newly infected individuals among 87 no exposures (0.184).

### Modeling the antibody response

To model the anti-N antibody response in individuals diagnosed with the first infection and re-infection with SARS-CoV-2 (Fig. 1c, d), a Bayesian model was used based on a previous study[28]. The measurement (i.e., $\log_{10}$ anti-N) $y_i$ was modeled as

$$y_i \sim Normal(h\,f(t_i, \alpha, \beta)\exp(-\lambda t_i), \sigma)$$

where *Normal(a, b)* indicates normal distribution with mean *a* and standard deviation *b*, $t_i$ is the time of the measurement, $f(t_i, \alpha, \beta)$ is the cumulative gamma distribution function at time $t_i$ with shape $\alpha$ and inverse scale $\beta$, $\lambda$ is the decay rate, $\sigma$ is standard deviation of the normal distribution, and *h* is the maximum response if $\lambda = 0$[28]. The posterior distribution of each parameter was sampled for each infection group. For the prior distribution of *h*, we used weakly informed priors, *Normal*(0, 5). For the prior distributions of $\alpha$, $\beta$, $\lambda$, and $\sigma$, we used a Student's t distribution with four degrees of freedom, instead of a normal distribution, to reduce the effects of outlier values[29].

Parameter estimation was performed using an MCMC approach implemented in *rstan*. Four independent MCMC chains were run with 4000 steps, including a warm-up period of 2000 steps, with subsampling at every five iterations. We confirmed that all estimated parameters showed <1.01 R-hat convergence diagnostic values and >1000 effective sampling size values, indicating that the MCMC runs were convergent. The estimated means are summarized in Supplementary Table 11.

### Statistical tests, correlation analysis, and data processing

The confidence intervals (CIs) of the categorical variables in the tables were calculated using two-sided Fisher's exact test. Pearson's correlation coefficients were used to assess correlations between continuous variables (Fig. 2c). In the correlation matrix analyses, Spearman correlations were calculated with false discovery rate (FDR) corrections (Fig. 2n–u), as described previously[30]. All statistical analyses were performed using R (version 4.3), and antibody titers below the detection limit were converted to half of the detection limit. Processed source data for anti-N antibody response model (Supplementary Data 1, 2), infection risk estimation (Supplementary Data 3), antibody responses (Supplementary Data 4), and correlation matrix (Supplementary Data 5) are listed in the Supplementary Data section.

### Reporting summary

Further information on research design is available in the Nature Portfolio Reporting Summary linked to this article.

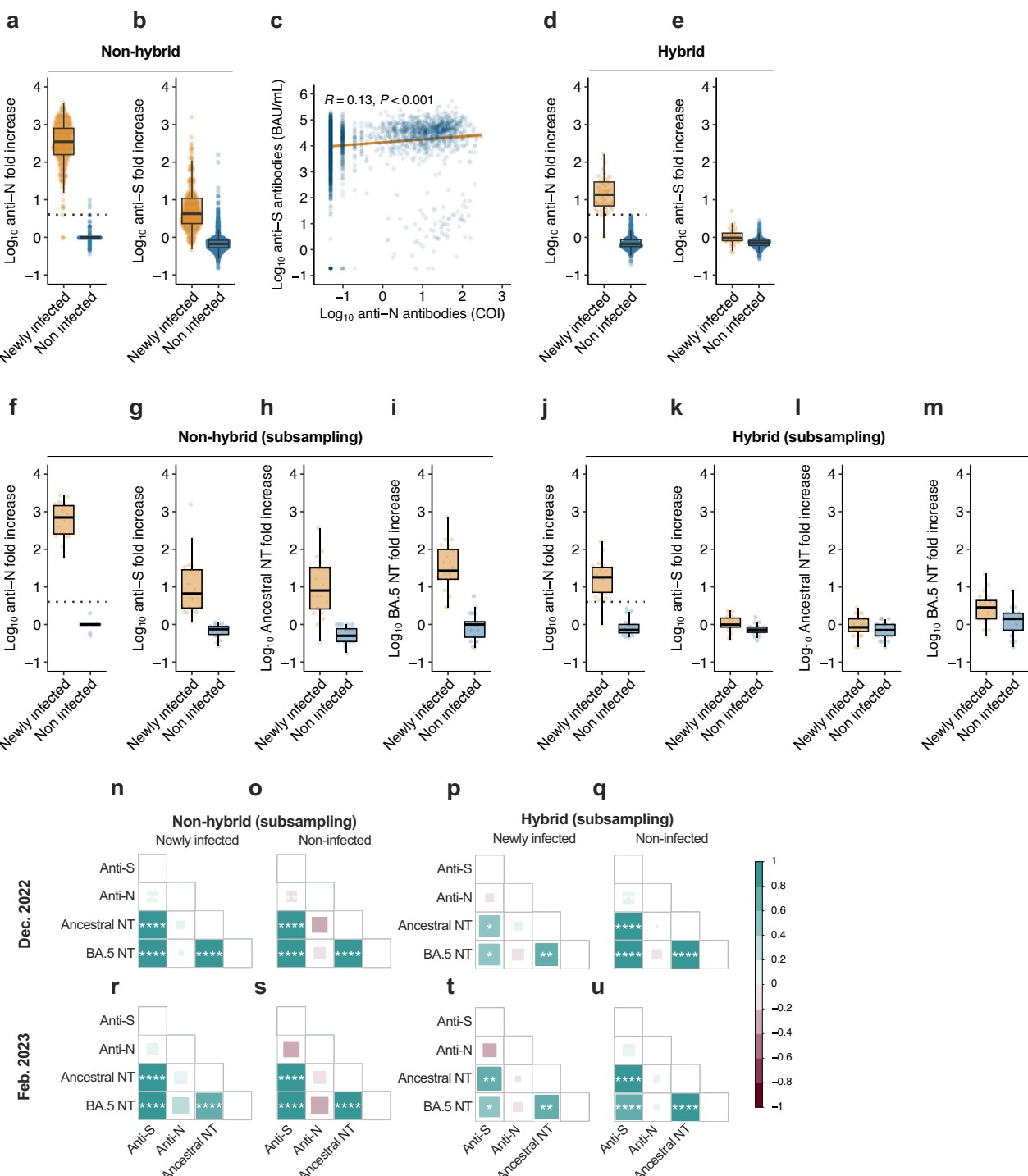

**Fig. 2 | Serum anti-S and anti-N antibody levels in the presence or absence of new infections. a, b** The antibody fold increase for participants without hybrid immunity from baseline (Dec. 2022) to the end of observation (Feb. 2023). **a** Anti-N antibody levels and **b** anti-S antibody titers. Each data point (dots) and the box plots are shown. The gray dotted line indicates the cutoff value (four-fold) based on Fig. 1d. $n = 3506$. **c** Correlation between serum anti-S and anti-N antibody levels for all participants. Data points (blue dots), correlation coefficient, ($P$ value, regression line, and 95% confidence interval (ribbon) are shown. $n = 4496$. **d, e** The antibody fold increases in individuals with hybrid immunity from baseline (Dec. 2022) to the end of observation (Feb. 2023). **d** Anti-N antibody levels and **e** anti-S antibody titers. Each data point (dots) and the box plots are shown. The gray dotted line indicates the cutoff value (four-fold) based on Fig. 1d. $n = 990$. **f–n** Antibody fold increase

before and after infection in subsampled specimens for neutralizing antibody response analysis. Anti-N antibody levels (**f, j**), anti-S antibody titers (**g, k**), and neutralizing antibody titers against the ancestral strain (**h, l**) and the BA.5 variant (**i, m**) are shown for individuals with (**f–i**) and without hybrid immunity (**j–m**), with and without new infections, at the time of the December 2022 survey and the February 2023 survey. Each data point (dots) and the box plots are shown. The gray dotted line indicates the cutoff value (four-fold) based on Fig. 1d. $n = 80$ participants. **n–u** Spearman correlation matrix of antibody titers in the subsampled specimens. The square size and a heat scale indicate Spearman correlation coefficients. The statistical significance level, corrected using the false discovery rate (FDR), is shown in the square; *$p < 0.05$, **$p < 0.01$, ***$p < 0.001$, ****$p < 0.0001$. $n = 80$ participants.

**Table 1 | COVID-19 seroprevalence during the study period**

| | | Overall (N = 4496) n (%) | 95% CI | Non-hybrid (N = 3506) n (%) | 95% CI | Hybrid (N = 990) n (%) | 95% CI |
|---|---|---|---|---|---|---|---|
| Dec. 2022 | Anti-S | 4400 (97.9%) | 97.4%, 98.3% | 3410 (97.3%) | 96.7%, 97.8% | 990 (100.0%) | 99.6%, 100.0% |
| | Anti-N | 1039 (23.1%) | 21.9%, 24.4% | 73 (2.1%) | 1.6%, 2.6% | 966 (97.6%) | 96.4%, 98.4% |
| Feb. 2023 | Anti-S | 4416 (98.2%) | 97.8%, 98.6% | 3426 (97.7%) | 97.2%, 98.2% | 990 (100.0%) | 99.6%, 100.0% |
| | Anti-N | 1358 (30.2%) | 28.9%, 31.6% | 413 (11.8%) | 10.7%, 12.9% | 945 (95.5%) | 94.0%, 96.7% |

## Results

### Characteristics of the study participants

Of the 15,000 invitees for the national COVID-19 seroepidemiological survey conducted as a public health investigation, 8157 participated in the December 2022 survey and 5627 participated in the February 2023 survey (response rate: 37.5%; Fig. 1a, b). We targeted the 5627 individuals who participated in both surveys. Those who received any COVID-19 vaccine during the observation period were excluded (1044 individuals) to eliminate the influence of the vaccination immune response during the study period. In addition, 87 individuals with a history of infection within 30 days before the initial test date were excluded (87 individuals) because the antibody response up to 30 days after infection is dynamic and difficult to accurately assess at this blood collection interval. Therefore, 4496 individuals were enrolled for the ad hoc study. Additionally, individuals with hybrid immunity were defined as vaccinated individuals with a history of prior diagnoses of COVID-19 or the anti-N antibodies positive (anti-N antibody levels ≥1.0; cutoff index, [COI]) at the initial test. The prevalence of anti-S antibodies in the five prefectures in Japan was previously reported at 97.2–98.3% in the December 2022 survey and 98.1–98.8% in the February–March 2023 survey[31,32]. The prevalence of anti-N antibodies was reported at 17.6–28.8% in the December 2022 survey and 22.6–35.8% in the February 2023 survey[31,32]. Detailed seroprevalence has been reported in Japan in the 2021–2022 and 2023 surveys[26,33]. The prevalence of anti-S and anti-N antibodies in the enrolled participants is shown in Table 1. The prevalence of anti-N antibodies in the hybrid immunity holder was ≥95% in both surveys. The prevalence of anti-N antibodies in the non-hybrid immunity holder increased from 2.6% in the December 2022 survey to 11.8% in the February–March 2023 survey.

We assessed the dynamics of anti-N antibody levels in individuals diagnosed with primary infection or re-infection to understand the differences in anti-N antibody responses and define the increase in anti-N antibody in SARS-CoV-2 re-infection. We applied a statistical model to assess the anti-N antibody response dynamics after diagnosis for all primary infection and re-infection cases using data obtained from cases with only the listed diagnosis date (primary infection, $n = 1218$; re-infection, $n = 26$) (Fig. 1c). For primary infection, anti-N titers were estimated to peak 69 days post-diagnosis and drop below the positive threshold (1.0 COI) at 621 days (95% credible interval (CredI), 522–754). For re-infection, the peak level of anti-N-antibody was estimated as 4.8 times higher than that of primary infection, and the duration of antibody levels above 1.0 COI was estimated to be longer than that of primary infection. Using this model, we calculated the fold increase in anti-N antibody levels from the pre-re-infection level to the level until 200 days post-re-infection (Fig. 1d). Since the interval from the initial to the second test in this cohort was approximately two months, the lowest fold increase in anti-N antibody level from the pre-re-infection state to 60 days after re-infection was estimated to be four-fold or higher. These anti-N antibody response dynamics in those who were re-infected were consistent with previous reported data using the same anti-N antibody detection kit[34,35]. These results suggest that a four-fold increase in serum anti-N antibody levels from the initial test to the second test in this cohort was considered a re-infection during the observation period.

For infection risk estimation, participants who tested positive for SARS-CoV-2 during the observation period, who were diagnosed with COVID-19 during the observation period, or who had a seroconverted anti-N titer from the initial to the second test (positive threshold, 1.0 COI) were considered newly infected cases during the observation period (Fig. 1b). Based on the anti-N antibody response model (Fig. 1c, d), participants who showed a four-fold or higher increase in anti-N antibody levels at the second antibody test compared to those at the initial test (baseline antibody level) were considered newly infected cases, including those with re-infections (Fig. 1b). The characteristics of newly infected and non-infected cases are presented in Supplementary Table 1, and the characteristics of newly infected cases with or without hybrid immunity before the study period are shown in Supplementary Table 2. Characteristics of individuals with symptomatic and/or asymptomatic infection during the study period are shown in Supplementary Table 3. Among the newly infected cases, the proportions of elderly individuals aged over 70 years, those with comorbidities, individuals who had received five vaccine doses, those who had received the Omicron-adapted bivalent vaccine, and individuals with a history of infection were relatively low (Supplementary Table 1). No apparent differences were observed between patients with or without hybrid immunity regarding comorbidity or vaccination status (Supplementary Table 2). Of the patients without hybrid immunity, 51.0% were diagnosed with COVID-19 during the study period. In contrast, only 5.6% of the re-infected patients with hybrid immunity were diagnosed with COVID-19, indicating that most of the re-infected patients with hybrid immunity were undiagnosed and undetectable without serological testing (Supplementary Table 2, 3). Similarly, only 5.6% of the re-infected cases with hybrid immunity were symptomatic, suggesting that most re-infected cases were asymptomatic and less likely to be recognized by symptom-based case identification (Supplementary Table 2).

### Antibody responses in newly infected cases in individuals with hybrid and non-hybrid immunity

In individuals with non-hybrid immunity, the median fold increase in anti-N antibody levels from the initial to the second test in newly infected cases with no hybrid immunity at the initial test date was 288.3 (IQR, 150.0–784.0; Fig. 2a, Supplementary Fig. 1a, b). In contrast, anti-S antibody titers in newly infected cases with no hybrid immunity at the initial test showed a limited increase (5.6-fold increase; IQR, 2.3–10.9; Fig. 2b) because the baseline level of anti-S antibody titers in this cohort was comparable with that of the booster vaccination due to high booster vaccination rate (71.2%; Supplementary Table 1) (Supplementary Fig. 1c, d). Additionally, the correlation between the anti-S and anti-N antibody levels among all participants was low (Fig. 2c). This suggests that the anti-N antibody response induced by infection during the Omicron BA.5 endemic period is significantly larger and superior for the detection of infection compared to the anti-S antibody response induced by infection in this cohort.

In individuals with hybrid immunity, the anti-N antibody levels of newly infected cases were lower in the initial test but higher in the second test than those in non-infected cases (Supplementary Fig. 1e, f). The median anti-N antibody levels' fold increase from the initial to second tests in cases with COVID-19 diagnosis during the study period was 14.0 (IQR, 6.8–29.6) for those with hybrid immunity before the study (Fig. 2d). In contrast, among newly infected cases with hybrid immunity, no clear increase in anti-S antibody titers from the initial to the second tests was observed

(Supplementary Fig. 1g, h), and anti-S antibody titers in cases with COVID-19 diagnosis during the study period hardly changed from the initial to second tests (1.1-fold increase; IQR, 0.8–1.3; Fig. 2e). Taken together, these results justify the use of the criteria of a four-fold increase in anti-N antibody levels during the study period to identify newly infected cases with or without hybrid immunity before the study period, including undiagnosed cases. For evaluating neutralizing antibody titers to the ancestral strain and BA.5 at the initial and second tests, we randomly selected 20 newly infected individuals with hybrid immunity and 20 without hybrid immunity (Fig. 1b). Furthermore, we performed propensity score matching for these 20 individuals to subsample 20 non-infected individuals during the observation period for each group (Supplementary Table 4). The neutralization assay with selected individuals revealed that the neutralizing antibody titers against the ancestral strain and BA.5 variant at the initial test showed indistinguishable differences between newly infected and non-infected individuals with hybrid immunity, similar to the anti-S antibody titer (Supplementary Fig. 1i–p). Furthermore, in individuals without hybrid immunity, the geometric mean fold increase in neutralizing antibody titers against the ancestral strain and BA.5 variant from the initial to the second testing was 9.8- and 32.6-fold, respectively (Fig. 2f–i, Supplementary Fig. 2). In contrast, individuals with hybrid immunity demonstrated that the increase in neutralizing antibody titers against the ancestral strain and BA.5 from the initial to the second tests for individuals with hybrid immunity was comparatively smaller (0.9-fold to the ancestral strain, and 2.7-fold to BA.5) than that observed in individuals with non-hybrid immunity (Fig. 2j–m, Supplementary Fig. 2). Next, the correlations between antibody titers in hybrid and individuals with non-hybrid immunity at each time point were evaluated (Fig. 2n–u). At any time point, the anti-S antibody titer, the neutralizing antibody titer against the ancestral strain, and the neutralizing antibody titer against BA.5 variant were positively correlated with each other in individuals with hybrid and non-hybrid immunity, while anti-N antibody levels did not correlate with any other antibody titers, suggesting that anti-N antibody levels serve as a marker of antiviral immunity, independent of neutralizing antibody titers.

### Estimating the infection risk reduction based on anti-S and anti-N antibody levels

The absolute risk reduction in newly infected individuals with or without hybrid immunity during the study period, which was the Omicron BA.5 endemic period, by combined serum anti-S and anti-N antibody levels at the initial test, was estimated using a generalized additive model with inverse probability weighting (Fig. 3a–i). Conditional effects of the anti-N antibody levels indicated that the risk of infection decreased logarithmically with increasing anti-N antibody levels in infection regardless of symptoms and symptomatic and asymptomatic infections (Fig. 3a, d, g). In symptomatic and asymptomatic infections, individuals with more than 1.0 COI and 17.2 anti-N antibody levels had a 90% relative risk reduction of new infections compared with a control group with no vaccination or prior infection history, respectively (Table 2). As shown above, anti-N antibody levels peaked 1–2 months post-infection and declined over time (Fig. 1c). The estimated median duration for which anti-N antibody levels remained over 17.2 COI preventing for asymptomatic infection was 117 (CredI, 99–132) days after primary infection and 505 (CredI, 276–>1000) days after re-infection, which is markedly shorter than the duration in the threshold 1.0 COI preventing for symptomatic infection as described above (Fig. 1c). While higher anti-S antibody titers were associated with a decreased risk of infection, the impact of this relationship was modest (Fig. 3b, e, h). In the risk estimation for symptomatic infection only, the threshold for 90% protection was estimated at 10,209 BAU/mL (Fig. 3e). Even at the highest observed anti-S antibody titer of 523,000 BAU/mL, the estimated reduction in relative risk was only 70% for asymptomatic infection (Fig. 3h). Evaluation of the combined impact of anti-N and anti-S antibody levels revealed a modest decrease in the absolute risk of infection attributable to anti-S antibody titers, whereas a reduction in the absolute risk of

infection was significantly associated with an increase in anti-N antibody levels (Fig. 3c, f, i).

Each antibody level at the time of the initial test was assessed to determine the impact of vaccine dosage and previous infection history before the study period. Anti-N antibody levels at the time of the initial test in patients with a prior infection history were not affected by variations in the vaccination dosage (Fig. 3j). In contrast, the anti-S antibody titers increased as the number of vaccinations increased, and individuals who were previously infected before the study period tended to have greater anti-S antibody titers than those vaccinated, regardless of the number or the type of vaccinations (Fig. 3k, l). These results indicate that the reduced risk of infection at high anti-S antibody titers was associated with infection-induced immunity and correlated more strongly with anti-N antibody levels than with vaccine-induced anti-ancestral S antibody titers.

### Estimating infection risk reduction based on BA.5 neutralizing antibody titers and anti-N antibody levels

To estimate the risk of infection using neutralizing antibody titers (NT) against BA.5, we performed a stratified random sampling of 7–9% of newly infected and non-infected individuals during the observation period, both for those with and without hybrid immunity (N = 340, Fig. 1b, Supplementary Table 5). Using the subsampled population, we estimated infection risk based on anti-S antibody titers and anti-N antibody levels. The estimated infection risks derived from the antibody titers in the subsampled population were consistent with those observed in the overall cohort for both symptomatic and asymptomatic infections, indicating that the subsampling had a low effect on estimating infection risk based on antibody levels (Supplementary Fig. 3). Subsequently, infection risks were estimated using BA.5 NT and anti-N antibody levels in the subsampled population (Fig. 4). The conditional effects of anti-N antibody levels indicated that the risk of infection decreased with increasing anti-N antibody levels for both symptomatic and asymptomatic infections, similar to the estimations based on anti-S antibody titers and anti-N antibody levels, shown in Fig. 3a–c (Fig. 4a–c, d, g). However, the conditional effects of BA.5 NT exhibited distinct patterns depending on symptom onset (Fig. 4b, e, h). For asymptomatic infections, the association between BA.5 NT and reduced infection risk was weak or negligible (Fig. 4h, i). In contrast, for symptomatic infections, the risk of infection clearly decreased with increasing BA.5 NT, and a substantial reduction in infection risk was observed when both anti-N antibody levels and BA.5 NT were high (Fig. 4e, f). For risk estimation of symptomatic infection only, the threshold for 90% protection was 10 BA.5 NT (Table 2). These findings indicate that serum anti-N antibody levels were strongly associated with reductions in the risk of both symptomatic and asymptomatic infections. However, anti-S antibody titers against the ancestral strain and neutralizing antibody titers against the circulating virus were primarily associated with reducing only symptomatic infection risk.

### Estimating the infection risk reduction in individuals with hybrid immunity

Finally, we estimated the absolute risk reduction for new infections, including only those with hybrid immunity, during the study period (Fig. 5). The conditional effect of anti-N antibody levels indicated that the risk of re-infection decreased on a logarithmic scale with an increase in anti-N antibody levels, as shown in Figs. 3a and 4a (Fig. 5a, c). However, the conditional effect of anti-S antibody titers or the combined effect with anti-N antibody levels did not show a link between high anti-S antibody titers and a reduced risk of re-infection (Fig. 5a–c). Similarly, the conditional effect of BA.5 NT or the combined effect with anti-N antibody levels did not show an association between high BA.5 NT and a reduced risk of re-infection (Fig. 5d–f). Given that anti-N antibody levels reflect the level of the immune response induced after viral infection[30], the level of serum anti-N antibodies determined

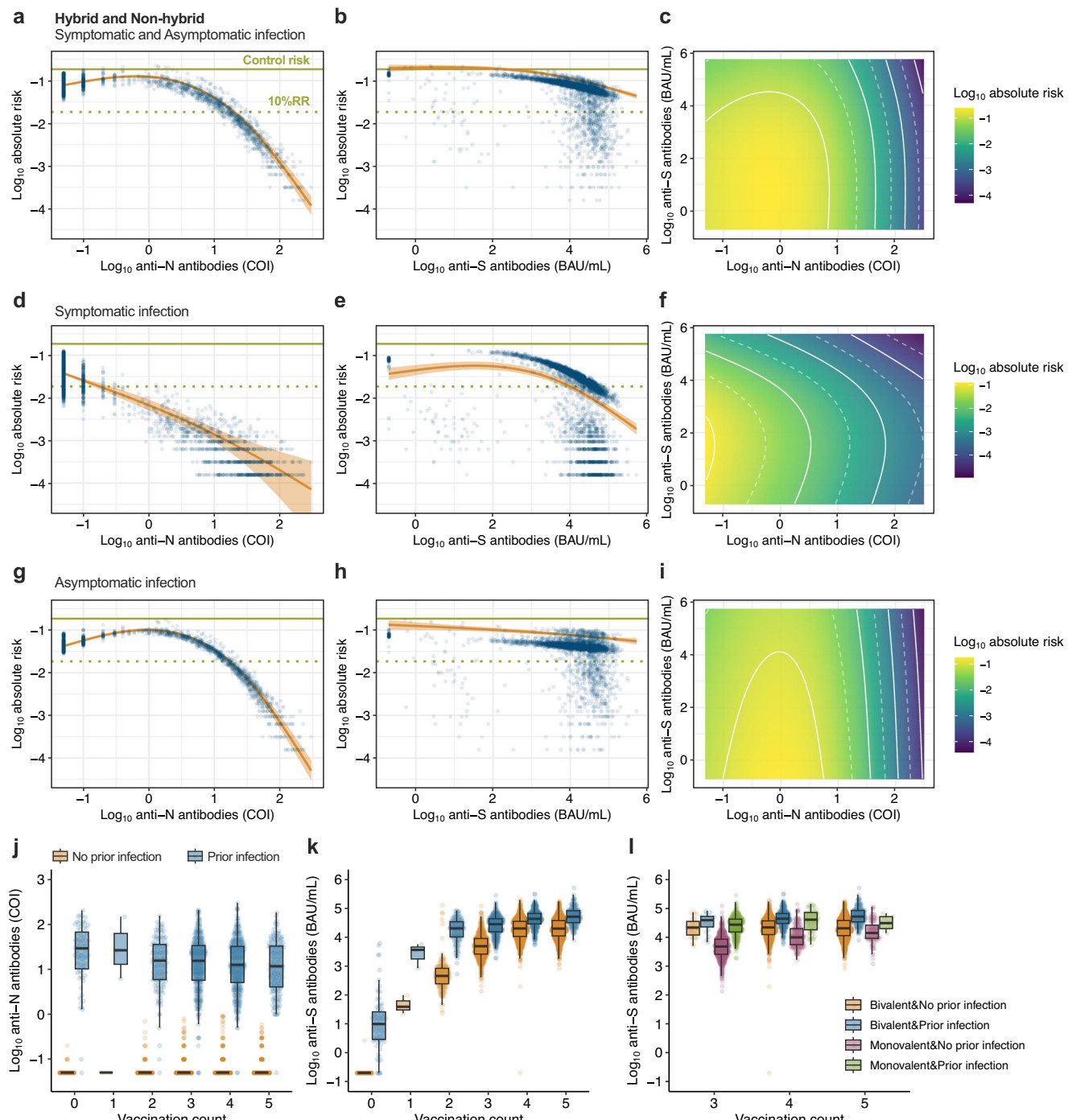

**Fig. 3 | Estimation of the associations of baseline anti-N and anti-S antibody levels on infection risk during the study period. a, b, d, e, g, h** The conditional effects of anti-N antibody levels and anti-S antibody titers during the observation period on the absolute risk of both symptomatic and asymptomatic infection (**a, b**), symptomatic infection (**d, e**), and asymptomatic infection (**g, h**), respectively. Each predicted data point (blue dots), along with the median absolute risk (line) and 95% CredIs (ribbon), is shown. The overall risk of a control group with no vaccination or prior infection history in this cohort (0.184) (green solid line) and the 90% reduction of relative risk (i.e., 10% relative risk [10% RR]) (green dotted line) are shown. **c, f, i** The combined effect of anti-N and anti-S antibody levels during

the observation period on the absolute risk of both symptomatic and asymptomatic infection (**c**), symptomatic infection (**f**), and asymptomatic infection (**i**), respectively. The logarithmic absolute risk of infection is indicated by the color bar. The white dotted and solid lines show the $\log_{10}$ absolute risk decrease for every 0.5 and 1.0, respectively. **j, k** Baseline anti-N antibody levels and anti-S antibody titers by the number of vaccine doses and infection history. **l** Baseline anti-S antibody titers by the number of vaccine doses, Omicron-adapted bivalent vaccination, and infection history. Each data point (dots) and the box plots are shown. Overall (**a–c, j–l**) and symptomatic infection (**d–f**); $n = 4496$. Asymptomatic infection (**g–i**); $n = 4327$.

using this method is a more reliable immunological correlate than the level of anti-S antibodies and NTs in determining the effectiveness of preventing SARS-CoV-2 re-infection in individuals with hybrid immunity.

## Discussion
In this study, we evaluated the protective effect of serum anti-N, anti-S, and Omicron BA.5 neutralizing antibody levels against SARS-CoV-2 infection during the Omicron BA.5 epidemic period from December 2022 to

**Table 2 | Antibody titer levels of 90% protection of SARS-CoV-2 infection (i.e., 10% relative risk) during the study period**

| | Overall cohort | | Subsampling | |
|---|---|---|---|---|
| | Anti-S (BAU/mL) Median (95% CredI) | Anti-N (COI) Median (95% CredI) | BA.5 NT Median (95% CredI) | Anti-N (COI) Median (95% CredI) |
| **Hybrid and Non-hybrid** | | | | |
| Symptomatic and Asymptomatic | >523,000 | 19.9 (18.2–21.9) | >3620 | 12.3 (8.5–18.9) |
| Symptomatic | 10,209 (6966–14,997) | <1.0 | 10 (3–38) | <1.0 |
| Asymptomatic | >523,000 | 17.2 (15.7–18.7) | >3620 | 11.7 (7.9–17.7) |
| **Hybrid** | | | | |
| Symptomatic and Asymptomatic | Undetermined | 18.1 (16.6–18.1) | Undetermined | 10.2 (6.2–20.2) |

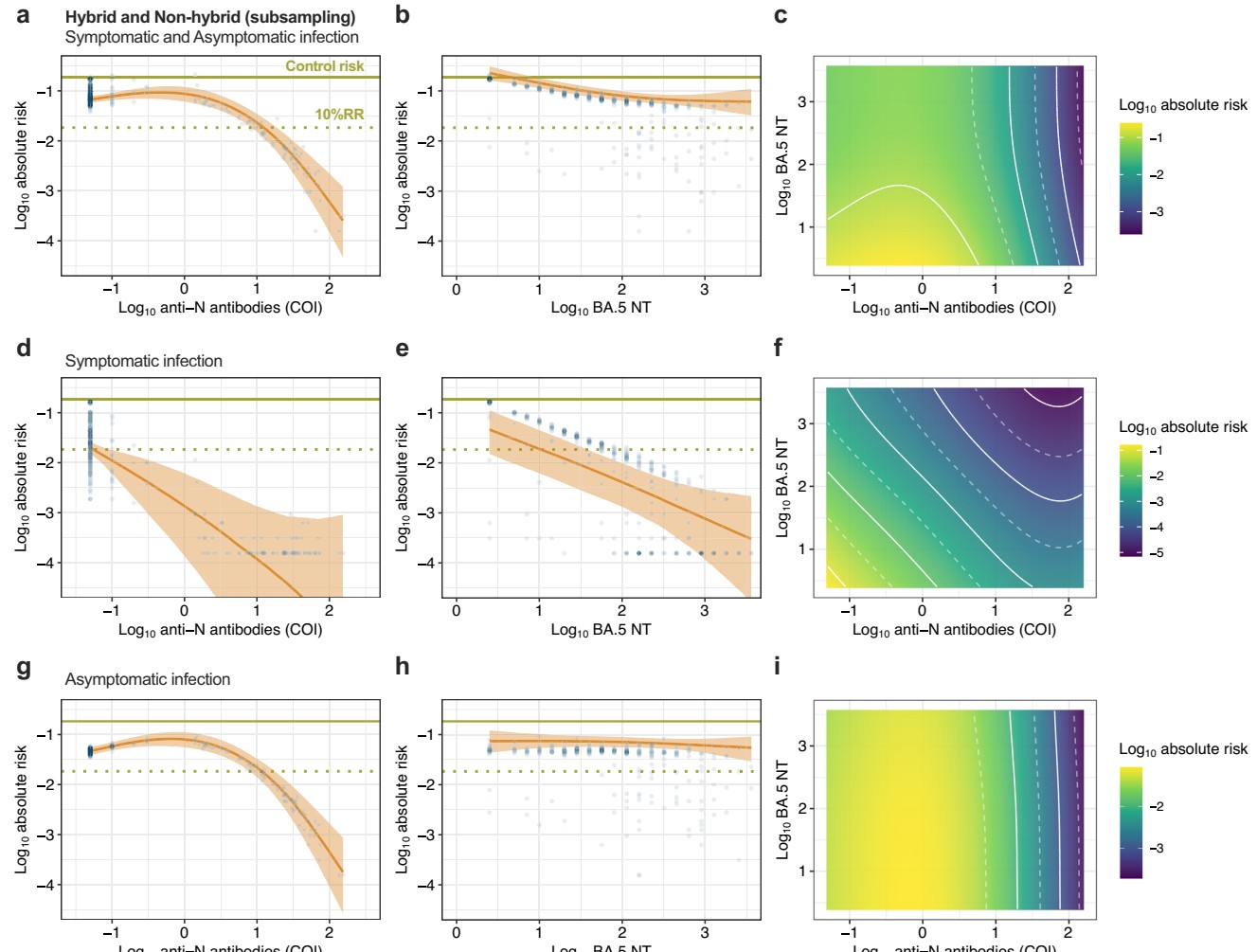

**Fig. 4 | Estimation of the associations of baseline anti-N and BA.5 neutralizing antibody levels on infection risk in the subsampling population. a**, **b**, **d**, **e**, **g**, **h** The conditional effects of anti-N antibody level and BA.5 neutralizing antibody titers (NT) during the observation period on the absolute risk of both symptomatic and asymptomatic infection (**a**, **b**), symptomatic infection (**d**, **e**), and asymptomatic infection (**g**, **h**), respectively. Each predicted data point (blue dots), along with the median absolute risk (line) and 95% CredIs (ribbon), is shown. The overall risk of a control group with no vaccination or prior infection history in this cohort (0.184) (green solid line) and the 90% reduction of relative risk (i.e., 10% relative risk [10% RR]) (green dotted line) are shown. **c**, **f**, **i** The combined effect of anti-N antibody level and BA.5 NT during the observation period on the absolute risk of both symptomatic and asymptomatic infection (**c**), symptomatic infection (**f**), and asymptomatic infection (**i**), respectively. The logarithmic absolute risk of infection is indicated by the color bar. The white dotted and solid lines show the $\log_{10}$ absolute risk decrease for every 0.5 and 1.0, respectively. Overall (**a–c**) and symptomatic infection (**d–f**); $n = 340$. Asymptomatic infection (**g–i**); $n = 324$.

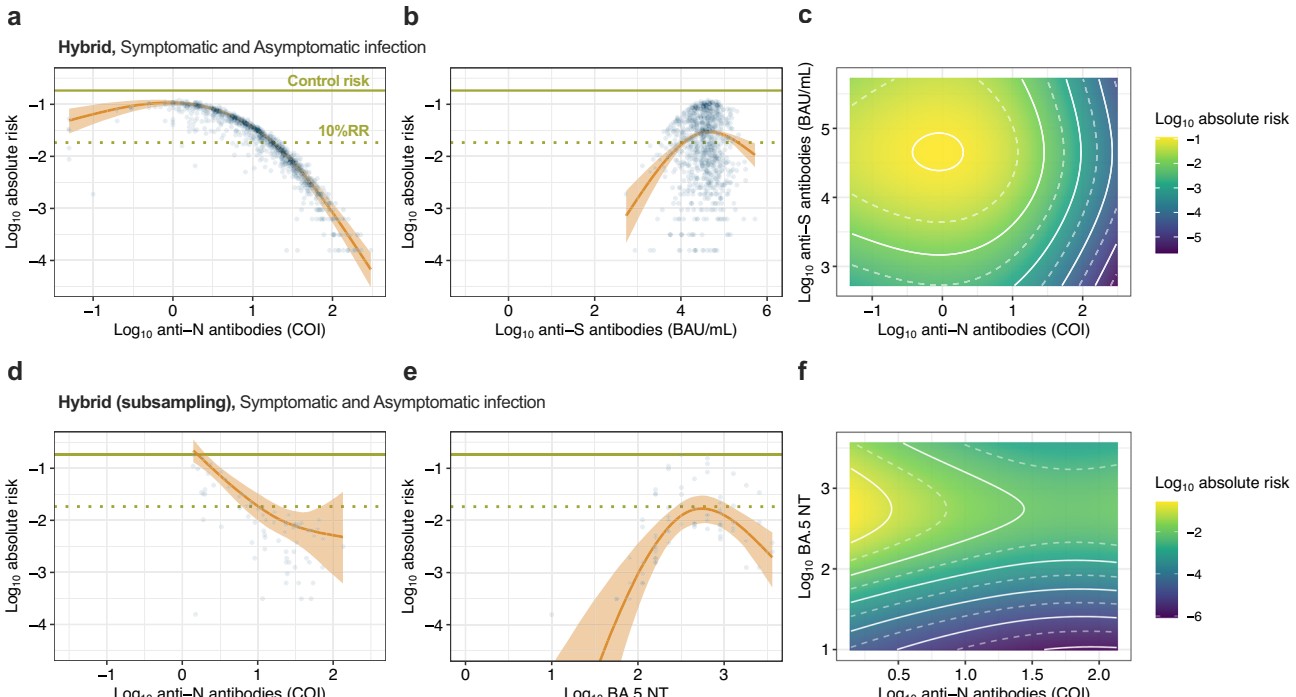

**Fig. 5 | Estimation of the associations of baseline antibody levels on re-infection risk in individuals with hybrid immunity. a, b** The conditional effects of anti-N antibody levels and anti-S antibody titers during the observation period on the absolute risk of re-infection in individuals with hybrid immunity at baseline. Each predicted data point (blue dots), along with the median absolute risk (line) and 95% CredIs (ribbon), is shown. The overall risk of a control group with no vaccination or prior infection history in this cohort (0.184) (green solid line) and the 90% reduction of relative risk (i.e., 10% relative risk [10%RR]) (green dotted line) are shown. **c** The combined effect of anti-N antibody levels and anti-S antibody titers during the observation period on the absolute risk of re-infection. The logarithmic absolute risk

of infection is indicated by the color bar. The white dotted and solid lines show the $\log_{10}$ absolute risk decrease for every 0.5 and 1.0, respectively. $n = 990$. **d, e** The conditional effects of anti-N antibody levels and BA.5 NT during the observation period on the absolute risk of re-infection in subsampled individuals with hybrid immunity at baseline. Each predicted data point (blue dots), along with the median absolute risk (line) and 95% CredIs (ribbon), is shown. **f** The combined effect of anti-N antibody level and BA.5 NT during the observation period on the absolute risk of re-infection. The logarithmic absolute risk of infection is indicated by the color bar. The white dotted and solid lines show the $\log_{10}$ absolute risk decrease for every 0.5 and 1.0, respectively. $n = 73$.

February 2023 in Japan. We showed that reducing symptomatic and asymptomatic infection risk was correlated with anti-N antibody levels at the initial test (December 2022), which were induced by previous infection before December 2022, on a logarithmic scale, even in individuals with hybrid immunity. In contrast, anti-S antibody titers against the ancestral spike antigen and BA.5 neutralizing antibody titers at the initial test (December 2022), which were induced by both vaccines and infections before December 2022, showed protective associations against only symptomatic infection during the BA.5 epidemic period. Interestingly, anti-S antibody titers against the ancestral spike antigen and BA.5 neutralizing antibody titers at the initial test exhibited no protective correlation against infection in individuals with hybrid immunity or against asymptomatic infection during the BA.5 epidemic period. These results suggest that the risk of re-infection can be assessed based on anti-N antibody titers in individuals with hybrid immunity. Moreover, this finding implies that infection-induced immunity in individuals with hybrid immunity includes a distinct and potent immunity beyond the presence of serum anti-S antibodies and neutralizing antibodies against the circulating strains, which correlates with serum anti-N antibody levels. This would indicate the possibility for the development of next-generation COVID-19 vaccines that induce more effective immunity.

Anti-N antibodies recognize the nucleocapsid proteins inside virions or infected cells and do not directly neutralize infectious virus particles, suggesting that anti-N antibody levels are non-mechanistic immunological correlates of protection (nCoPs), which are associated with some protective immunity against infection[36]. The potential of serum anti-N antibody levels to serve as immunological markers against re-infection has been reported in children aged 4–15 years in a UK pediatric cohort, which is consistent with

the findings of this study[37]. Immunological markers induced by infection, such as nasal mucosal S-specific secretory IgA antibodies[21], which are also associated with a reduction in the duration of virus shedding after infection[22] and blood circulating N-specific $CD4^+$ and $CD8^+$ T cells induced by infection, but not spike-based current vaccination, associated with the prevention of infection, have been reported to negatively correlate with peak upper respiratory tract viral loads[38]. Importantly, these immunological markers represent humoral and cellular immune responses to viral infections in the upper respiratory tract. Serum anti-N antibody responses are similarly considered to be indicative of an immune response to viral infections in the upper respiratory tract. Previously, we reported a positive correlation between the serum anti-N antibody response and upper respiratory viral load at the onset of breakthrough infections[30], and revealed that the upper respiratory viral load determines the magnitude of antiviral humoral immune responses after infection. Regarding T cell immunity, the non-hospitalized COVID-19 cases had higher $CXCR5^+$ $CD4^+$ T follicular helper cell responses than hospitalized cases, and virus-specific $CD4^+$ T cell responses correlated positively with serum anti-N and anti-S antibody levels within 180 days post-onset[39]. In breakthrough infections after vaccination, which the majority of individuals with hybrid immunity, an increased $CD4^+$ T cell response to various SARS-CoV-2 proteins, including N proteins, has been reported[40], suggesting that the enhanced $CD4^+$ T cell response against non-spike proteins may be reflected in higher serum anti-N antibody levels in individuals with hybrid immunity. Taken together, these previous findings indicate that serum anti-N antibody levels serve as a quantitative proxy for the magnitude of antiviral immune responses induced after infection.

The correlation between anti-nucleoprotein antibodies and a reduced risk of viral infection has not been reported in cases of influenza or other

respiratory viruses, emphasizing the novelty of our findings. In human influenza research, the importance of nucleoprotein-specific CD8[+] and CD4[+] T cells in protection against influenza virus infection has been emphasized[41,42]. Furthermore, studies on universal influenza vaccines targeting nucleoproteins OVX836 have shown suppression of symptomatic influenza infections through nucleoprotein-targeted immunity in Phase II clinical trials[43]. For SARS-CoV-2, vaccines combining S and N proteins can enhance immunogenicity and efficacy through CD8[+] T cell-dependent mechanisms in pre-clinical models[44]. Moreover, SARS-CoV-2 N proteins are expressed on the cell surface of infected cells, making it a potential target for anti-N antibodies mediated destruction of infected cells[45]. Similar phenomena have been reported with other RNA viruses, including the Influenza virus[46], where anti-nucleoprotein antibodies mediate lysis of virus-infected cells[47] and provide infection protection in mouse models[48,49]. For SARS-CoV-2 infection, in vitro studies have shown that cell surface N protein expression can induce antibody-dependent cellular cytotoxicity (ADCC) via the anti-N antibody Fc domain[45]. In addition, treatment with anti-N sera or monoclonal antibodies suppressed viral replication by inducing NK cell-mediated ADCC in mice models[50]. Considering these studies alongside our findings, it is plausible that immunity targeting nucleoprotein/nucleocapsid is strongly associated with controlling respiratory viral infections.

Our results showed that asymptomatic infections occurred more proportionally in newly infected individuals with hybrid immunity than those without hybrid immunity (Supplementary Table 2), suggesting that individuals with hybrid immunity possess robust protective mechanisms that effectively prevent the symptom onset. On the upper respiratory tract, antiviral mucosal secretory IgA antibodies are produced within a few days after onset[22]. This mucosal antibody response is thought to restrict the expansion of viral infection to the lower respiratory tract, potentially resulting in asymptomatic or mild infections[51]. It was also reported that nasal anti-S IgA responses after onset were correlated with lower viral loads and symptom resolution[52]. Our previous work also showed that re-infected individuals with prior infection mounted a faster and higher nasal secretory IgA response, which was associated with shorter periods of viral shedding[22]. However, our previous study showed that no correlation was observed between the early secretory IgA response and the symptom onset or duration of symptomatic period; thus, it remains unclear whether the mucosal antibody response directly links to the symptom prevention[22]. The current study did not collect any mucosal specimens, and the relationship between infection-induced mucosal immunity and symptom prevention remains unclear. Further research is needed to elucidate immunological factors involved in symptom prevention and their connection to mucosal immunity in individuals with hybrid immunity.

For individuals who experienced a single infection, the time for anti-N antibody levels to become negative (below 1.0 COI) was estimated to be approximately 621 days (Fig. 1c). These anti-N antibody waning dynamics are consistent with a recent longitudinal sampling cohort study using the same method of anti-N antibody measurement as in this study, which showed that many participants maintained positive anti-N antibody levels for more than 500 days[34,53]. In addition, a previous report found that anti-N antibody levels were higher in re-infections than in primary infection[35], which is consistent with the findings of this study. Notably, the anti-N antibody levels induced by the primary infection were relatively low and below a 90% risk reduction level for re-infection at 117 days after the primary infection (Fig. 1c). This suggests that a single infection does not induce robust long-term immunity for preventing re-infection. This is consistent with systematic reviews indicating that the effectiveness of preventing infection in previously infected individuals against re-infection with Omicron declines to approximately 50% within 20 weeks[20]. Nasal anti-S IgA has been reported to persist for approximately nine months, and strong correlations were observed between nasal anti-S IgA and nasal anti-N IgA titers at six or 12 months after infection, but not with plasma anti-N IgA titers[54]. Currently, the long-term relationship between serum anti-N antibodies and mucosal-specific secretory IgA or T cell immunity is unclear,

and further research is needed to identify the long-term immune responses associated with anti-N antibody levels.

Several studies conducted during the pre-Omicron epidemic period following vaccine introduction have reported associations between elevated anti-S antibody or neutralizing antibody titers and strong protection against SARS-CoV-2 infection[9,10]. Feng et al. demonstrated 80% protection against SARS-CoV-2 infection when the anti-S antibody titer was 247 BAU/ml in a ChAdOx1 study cohort. Gilbert et al. reported that mRNA-1273 vaccine efficacy increased with anti-S IgG levels; for anti-S IgG titers of 33, 300, and 4000 BAU/ml, the vaccine efficacies were 85% (31–92%), 90% (77–94%), and 94% (91–96%), respectively[4]. During the Omicron BA.1/BA.2 epidemic period, reductions in infection risk were reported for anti-S antibody titers above 2000 BAU/mL, 800 BAU/mL, and 4810–11233 BAU/mL for symptomatic infections[12–14]. Acute-phase ancestral anti-S antibody levels associated with 50% lower odds of COVID-19 were 1968 BAU/mL against Delta and 3375 BAU/mL against Omicron 1/2, showing that the required anti-S antibody levels for preventing symptomatic infection were increased for Omicron[16]. Additionally, during the Omicron BA.4/5 epidemic period, reductions in infection risk were reported for anti-S antibody titers above 380–1560 BAU/mL, but the anti-S antibody titer thresholds varied according to previous infection histories, and the association between the anti-S antibody titers and the infection risk was not observed in individuals with previous Omicron BA.2 infection[15]. Even in a study using neutralizing antibody titer for assessing the correlation with re-infection risk, the association between the anti-BA.1 neutralizing antibody titer in prior-infected individuals and protection against re-infection during the Omicron BA.1/2 period was limited to 11%[55]. Our results estimated 10209 BAU/mL of anti-S antibody titer, and 10 BA.5 NT had a 90% relative risk reduction against symptomatic infection. However, we did not find a 90% relative risk reduction for asymptomatic infection or infection in individuals with hybrid immunity, even at anti-S antibody titers above 100,000 BAU/mL and surpassing 3000 BA.5 NT during the Omicron BA.5 epidemic period in this study. This may indicate that the protection conferred by existing COVID-19 vaccines is insufficient to prevent asymptomatic infection with the currently circulating omicron sublineages.

Despite the overall strengths of this study, it has several limitations. First, the observation period of this study was limited to approximately two months; therefore, a longer observation period could provide a better estimation of the long-term protective effects and the precise duration of immunity for infection protection. Second, this study included only adult residents of Japan. Therefore, the findings in this study may not be generalizable to other populations with different demographics and vaccination statuses, other than populations in the UK pediatric cohort study[37], suggesting some consistency across different populations. Third, the estimated anti-N antibody increase after re-infection in this study represented the overall trend in the study population and did not account for individual variations in anti-N antibody responses. While the criterion with a four-fold increase in anti-N antibody levels, which our anti-N antibody dynamics model determined, is expected to detect the most re-infection cases in the study population, there is a possibility of failing to detect re-infection cases whose anti-N antibody titer has not increased four-fold. Fourth, for estimating the infection risk reduction based on antibody levels, the reliability of estimates in regions with sparse antibody titer data may be low. While cubic splines with limited basis functions (k = 3) were used to stabilize these estimates, this limitation should be considered when interpreting the results. Fifth, we did not evaluate T cell immunity and Fc-effector function in humoral immunity, which could play significant roles in protection against infection[16]. Sixth, we did not evaluate the effect of antibody levels on the prevention of severe disease since the questionnaire did not include the severity in this survey, although the prevention of severe disease through vaccination has been confirmed even during the Omicron BA.1/2 and BA.4/5 epidemic periods. Seventh, while the participants with a prior infection history included pre-Omicron-infected individuals, most were Omicron-infected cases. The quality of immunity induced by infection may vary with the variant, but it has not been determined which variant infected each

individual with a prior infection history. Eighth, we have not examined the correlation between serum anti-N antibody levels and prevention for the other variant infections outside the BA.5 epidemic period. Further investigation is needed to determine if the correlation between anti-N antibody titers and infection risk reduction can be applied to Omicron sublineages post-BA.5.

## Conclusion

In this study, we found that, during the Omicron BA.5 epidemic period, the reduction in re-infection risk was significantly correlated with higher anti-N antibody levels induced by prior infection. Conversely, anti-S antibody titers and BA.5 neutralizing antibody titers induced by both vaccines and infections were less strongly correlated with protection. These findings suggest that immunity correlated with anti-N antibody levels—such as mucosal antibodies, T cell responses, and other unknown factors—may be a good target for immunity induced by the next-generation COVID-19 vaccines aiming the control the future COVID-19 epidemic.

## Data availability

Source data for the main figures in the manuscript are provided in Supplementary Data files 1–5.

## Code availability

The R codes for infection risk estimation and modeling the antibody response are available in a Zenodo Repository[56].

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

## Acknowledgements

We thank the Miyagi, Tokyo, Aichi, Osaka, and Fukuoka prefecture governments for their support in implementing the national COVID-19 seroepidemiological survey. We also thank Shoko Sakuraba and Jun Sugihara for their support in implementing the survey from the Ministry of Health, Labour and Welfare, Japan. We also thank the staff members at the Survey Research Center, Mitsubishi Research Institute, SRL, Inc., and Benefit One Inc. for their administrative and technical assistance with the survey. We thank Emi Taeda at the Department of Pathology, National Institute of Infectious Diseases, for her technical assistance. The national COVID-19 seroepidemiological survey was funded by the MHLW as a public health investigation. The MHLW funded and was involved in the survey design and selection of the participants for the national COVID-19 seroepidemiological survey to estimate SARS-CoV-2 seroprevalence in Japan as a public health investigation. The ad hoc study based on the survey data as a research activity was supported in part by a Grant-in-Aid for JSPS Scientific Research (KAKENHI) (23K27422 (to T.S.), 21K20768 (to S.M.), 23K14534 (to S.M.), and 21K17307 (to R.K.)); AMED Research Program on Emerging and Reemerging Infectious Diseases (JP24fk0108637 (to T.S.), JP22fk0108509 (to T.S.), JP25fk0108684 (to T.S.), JP25fk0108732 (to T.S.), and JP22fk0108568 (to S.M.)); MHLW Emerging/Reemerging Infectious Diseases and Vaccination Policy Promotion Research Project (24HA2009 (to T.S.), 22HA2006 (to T.S.), 21HA2005 (to T.S.) and 20HA2001 (to T.S.)); the JST, PRESTO (JPMJPR21RC (to D.Y.)).

## Authors contributions

Conceptualization, S.M., R.K., T.A., H.T., H.H., S.A., M.S., D.Y., T.W., and T.S.; Methodology, S.M., R.K., T.A., D.Y., and T.S.; Research investigation, S.M., K.N., R.K., T.A., M.H., R.S., M.S., D.Y., T.W., and T.S.; Public health investigation, S.M., R.K., H.T., H.H., T.K., A.S., A.A., S.A., M.S., D.Y., T.W., and T.S.; Data curation, S.M., K.N., M.H., R.S., and T.S.; Computational analysis, S.M. and K.N.; Formal analysis, S.M., K.N., M.H., and T.S.; Visualization, S.M.; Research funding acquisition, S.M., R.K., M.S., D.Y., T.W., and T.S.; Project administration, M.S., and T.S.; Supervision, M.S., Y.D., and T.S.; Writing the original draft, S.M. and T.S.; Writing – review & editing, S.M., K.N., R.K., M.S., D.Y., and T.S. All authors agreed to submit the manuscript, read and approved the final draft, and take full responsibility for its content, including data accuracy and statistical analysis.

## Competing interests

The authors declare no competing interests.

## Additional information

[1]Department of Infectious Disease Pathology, National Institute of Infectious Diseases, Japan Institute for Health Security, Tokyo, Japan. [2]Department of Infectious Disease Pathobiology, Graduate School of Medicine, Chiba University, Chiba, Japan. [3]Department of Epidemiology, National Institute of Infectious Diseases, Japan Institute for Health Security, Tokyo, Japan. [4]Department of General Medicine, Juntendo University Faculty of Medicine, Tokyo, Japan. [5]Research and Development Coordination Office, National Center for Global Health and Medicine, Japan Institute for Health Security, Tokyo, Japan. [6]Department of Immunization Research, National Institute of Infectious Diseases, Japan Institute for Health Security, Tokyo, Japan. [7]Center for Infectious Disease Epidemiology, National Institute of Infectious Diseases, Japan Institute for Health Security, Tokyo, Japan. [8]Japan Institute for Health Security, Tokyo, Japan. ✉e-mail: tksuzuki@niid.go.jp

