## [Transparent Peer Review file · Communications Medicine]

Serum anti-nucleocapsid antibody correlates of protection from SARS-CoV-2 re-infection regardless of symptoms or immune history

Corresponding Author: Dr Tadaki Suzuki

Version 0:

Reviewer comments:

Reviewer #1

(Remarks to the Author)

The manuscript titled "Serum anti-nucleocapsid antibody level is a correlate of protection against SARS-CoV-2 re-infection in hybrid immunity holders" explores the correlation between anti-nucleocapsid (N) antibody levels and protection against SARS-CoV-2 re-infection in individuals with hybrid immunity (those who have been both vaccinated and previously infected). The manuscript provides valuable insights into the role of anti-N antibodies in protection against SARS-CoV-2 re-infection in hybrid immunity holders. However, it would benefit from addressing the comments mentioned below to enhance its robustness and generalizability. The findings highlight the complexity of immune protection against SARS-CoV-2 and underscore the need for continued research and development of more effective vaccines. I recommend major revision.

Minor comments/concerns:

- The study is limited to Japanese adults, which may not be generalizable to other populations with different demographics and vaccination statuses.
- Excluding individuals who received any COVID-19 vaccine during the observation period may affect the generalizability of the findings to real-world scenarios where booster doses are common.
- The reliance on antibody levels as a sole measure of immunity can be limiting. T-cell responses and other aspects of the immune system are not considered, which could play significant roles in protection.
- The study uses a four-fold increase in anti-N antibody levels as a criteria for identifying re-infections, which could be arbitrary and might not account for individual variations in immune responses.

Major comments:

- All samples from the cohort (non- and hybrid immunity individuals) were tested for anti-S and anti-N, but not for live or pseudovirus neutralization (NT50). Although anti-S and anti-N were expressed as a cutoff index, it would be crucial to report the IC50 titers from nAb, which would provide estimated values for 50% of protection. It would be nice to have a comparison of the antibody levels and nAb titers pre-reinfection for cases with control sample taken at a similar period of time. It would be nice to compare neutralization titers and their longitudinal dynamics in individuals with hybrid immunity or not.
 - Authors do not analyze trajectories of antibody levels and neutralization titers before and after reinfection.
 - Authors should show the correlation between neutralization assays and antibody levels (Anti-S and anti-N).
 - Global SARS-CoV-2 seroprevalence has been observed to have substantial variation in the proportion of immunity induced by infection or vaccination in different settings worldwide (Bergeri I et. al. published). Authors should describe the seroprevalence for both hybrid immunity patients and non hybrid immunity.
 - The duration of the antibody response against SARS-CoV-2 differs broadly across individuals. Therefore, it is important to measure the peak of antibody titers at baseline and over time. It is possible that the intensity of the initial antibody response allows the estimation of long term antibody duration.
- The discussion should be more elaborate, including other studies pre-omicron or after Omicron (newly variants), in which anti-S showed protection.

Reviewer #2

(Remarks to the Author)

Overall comments

1. A strength of the study is that it was done over a short period of time in the known Omicron BA.5 epidemic.
2. A slight weakness is only measuring antibodies to the Ancestral strain instead of also to Omicron BA.5 that would enable

studying variant-matched correlates. I believe it is only a slight weakness given literature supporting that Ancestral strain antibodies tend to be as good as correlate as variant-matched antibodies.

3. Multiple infectious disease outcomes are studied such as new symptomatic COVID-19, new asymptomatic infection, and new infection regardless of symptoms. I had a hard time finding the numbers of endpoints/outcomes of the different types. It may strengthen the article to add more dissection/description of the numbers of different outcomes (stratified by key covariate groups). This issue surfaced especially at page 11, lines 305-306.

4. The manuscript supports the importance of integrating anti-N concentration data in immune correlates analyses of COVID-19 vaccines in populations that are largely previously infected. Could the reviewers put this result in the context of flu vaccine research (alternatively for other relevant pathogens) where immune correlates are also studied in previously infected populations? It would be valuable to communicate whether the main messages of this article regarding the improvement of immune correlates through including anti-core protein antibody responses (non-vaccine antigens) in addition to surface Envelope vaccine protein antibody responses carries over to other pathogens, and whether there is a history of this being effectively done.

5. The manuscript often reports on 'infection prevention efficacy' which is presumably efficacy from hybrid immunity (vaccination and prior infection) vs. vaccine immunity only (without prior infection), but I found the descriptions confusing at times. I'm not sure what 'infection prevention efficacy' refers to, where it could refer to the protection from a recent infection regardless of vaccination pattern. In general protection depends on both previous infection history and vaccination history, and I didn't think the way in which both of these factors were accounted for was sufficiently clearly described. Another way to say this comment, is I'm not sure that vaccination history information was accounted for as completely as may be helpful.

6. Tied to the last comment, adjustment for vaccination status and vaccine type is relevant for confounding control and effect modification assessment; can more details be added for how the vaccination history variables are studied?

7. In the Statistical analysis section on page 12, use of logistic regression is described. However, elsewhere the manuscript describes that a Bayesian GAM and Poisson regression are also applied. The coordinated use of these 3 approaches is not described, ideally there would be a mapping of which method is applied to which objective and why, so the reader could easily see the rationale for this set of methods for meeting all of the objectives. It looks like this 'Statistical analysis' heading is meant to fit under 'Modeling the antibody response', but the way it is written it comes across as if it is the consolidated section describing statistical analyses.

Specific comments:

1. Page 3, lines 52-54, "Additionally, recent reports..." Clarify this finding?

2. Page 3, lines 69-70, I think what is meant is more like "association of ... antibody levels with the occurrence of a new infection."

3. Page 4, The data contribute information about the value of anti-N concentration for detecting new infections. Can the authors comment on how well anti-N concentration can be used to estimate the timing of infection?

4. Page 5, lines 134-135: 'infection prevention efficacy' seems like a vague term, where just after this section heading it is explained that the efficacy of hybrid immunity vs. not hybrid immunity is clearer. Although this phraseology is not completely clear either, because it is isn't clear whether hybrid immunity is being compared to vaccinated not previously infected or to not vaccinated and not previously infected, or to a mixture (not previously infected regardless of vaccination history).

5. I think the first paragraph of the Discussion would be improved if it includes information on the timing of sample draws relative to outcomes, otherwise it is too vague of a summary.

6. Page 7, line 186, I think 'non-mechanistic' is meant, not 'non-mechanical' ('non-mechanistic' is the term in the literature).

7. Page 11, lines 305-306, and elsewhere in Methods. Aspects of the writing such as 'turn positive' is too imprecise, please explicitly write that the diagnosis is based on such and such data from information available at the second visit. Relatedly, in the Infection risk estimation section, lines 319-324, the description omits the information on time points at which antibody markers are measured, and/or case status is ascertained. As this kind of information is basic to the analysis, it should be directly described explicitly, rather than the way it is done now leaving some details tacit that the reader needs to infer. E.g., when it says the Poisson model is used to predict the probability of infection, it would be useful to state diagnosis of infection over what temporal period and diagnosed at what time point, etc.

8. Page 11, lines 329-330. It is not clear how weighing the correlates of risk models eliminates bias, this needs more details.

9. Fig 3. Caption, "Estimation of the effect..." Consider changing the word 'effect' to 'association' given that antibody levels were not randomized and formal causal inference methodology was not employed that would enable pursuing the assessment of effects.

10. Fig. 3 shows very narrow confidence intervals, even in the right-tail of anti-N concentration where there is little data support. Therefore, this analysis seems to report more precision than is actually available. I think this is because of the

parametric models that are used and precision is quantified assuming the parametric shapes are correct. Consider adding to Limitations in the discussion.

Version 1:

Reviewer comments:

Reviewer #1

(Remarks to the Author)

I thank the authors for addressing most of my concerns.

While I understand that the role of T-cell responses and mucosal immunity is beyond the immediate scope of the paper, elaborating on their contribution to hybrid immunity could enhance the discussion. This addition would align the findings more closely with the broader immunological context and provide further insights into the observed correlations.

Specifically:

Discussing the interplay between T-cell responses and anti-N antibody levels could offer a more comprehensive understanding of the mechanisms behind hybrid immunity.

Addressing mucosal immunity could strengthen the paper's relevance, particularly given its established role in asymptomatic and upper respiratory tract infections.

Reviewer #2

(Remarks to the Author)

I appreciate the thorough response letter and revision, which has satisfactorily addressed my comments on the initial submission.

Point-by-Point Response to Reviewers' comments:

Thank you for providing invaluable feedback on our research. We have addressed each comment in a point-wise manner, indicated in blue text. Additionally, we included some additional figures and their legends in this response letter. Changes and additions to the main text have been highlighted in yellow for your ease of review.

Reviewer #1 (Remarks to the Author):

The manuscript titled "Serum anti-nucleocapsid antibody level is a correlate of protection against SARS-CoV-2 re-infection in hybrid immunity holders" explores the correlation between anti-nucleocapsid (N) antibody levels and protection against SARS-CoV-2 re-infection in individuals with hybrid immunity (those who have been both vaccinated and previously infected). The manuscript provides valuable insights into the role of anti-N antibodies in protection against SARS-CoV-2 re-infection in hybrid immunity holders. However, it would benefit from addressing the comments mentioned below to enhance its robustness and generalizability. The findings highlight the complexity of immune protection against SARS-CoV-2 and underscore the need for continued research and development of more effective vaccines. I recommend major revision.

Reply to comment:

We would like to thank the reviewer for their helpful comments and the high value they place on the findings presented in our manuscript, particularly the potential contribution of our findings to vaccine research and development. We believe that our additional analysis following the reviewers' suggestions markedly enhances the robustness and generalizability of our study.

Minor comments / concerns:

-The study is limited to Japanese adults, which may not be generalizable to other populations with different demographics and vaccination statuses.

Reply to comment:

As pointed out by the reviewer, the study population consisting only of Japanese adults is one of the limitations of our study. We have added the description of this limitation in the revised manuscript (Lines 325–328). Notably, a study from a UK pediatric cohort reported an association between anti-N antibody levels and protection against symptomatic reinfection during the Omicron BA.4/5 wave from January-May 2022 to October-December 2022, which has been stated in the discussion section (Lines 249–251).

Lines 249–251:

The potential of serum anti-N antibody levels to serve as immunological markers against re-infection has been reported in children aged 4 to 15 years in a UK pediatric cohort, which is consistent with the findings of this study.³⁴

Lines 325–328:

Therefore, the findings in this study may not be generalizable to other populations with different demographics and vaccination statuses, other than populations in the UK pediatric cohort study,³⁴ suggesting some consistency across different populations.

-Excluding individuals who received any COVID-19 vaccine during the observation period may affect the generalizability of the findings to real-world scenarios where booster doses are common.

Reply to comment:

Our study aimed to examine whether antibody levels at the start of the observation period (baseline antibody levels) were associated with a reduced risk of infection during the two-month observation period. COVID-19 vaccinations during this observation period can cause a considerable increase in anti-S antibody levels, which might contribute to reducing the risk of infection in these individuals. This could potentially distort the outcomes of our analysis, which assesses the risk of infection using antibody titers at the beginning of the observation period. Therefore, individuals who were vaccinated during the observation period were excluded. We added this description to the Methods.

Lines 382–385:

To fairly examine whether antibody levels at the beginning of the observation period (baseline antibody levels) were associated with a reduced risk of infection during the two-month observation period, those who were vaccinated during the observation period were excluded (1,044 individuals).

In addition, we analyzed 5,523 individuals without excluding those who received COVID-19 vaccinations during the observation period to address the reviewer's concern (**Additional Figure 1**). The results showed that the associations between baseline serum anti-N and anti-S antibody levels and reduced infection risk were consistent with those of the analysis in the revised manuscript, further supporting the robustness of our findings.

Additional Figure 1

Additional Figure 1. Estimation of the effect of baseline anti-N and anti-S antibody levels on infection risk during the study period in the population included additional vaccinated participants after the first antibody test.

(A, C, E) The conditional effects of anti-N antibody levels (left) and anti-S antibody titers (right) during the observation period on the absolute risk of both symptomatic and asymptomatic infection (A), symptomatic infection (C), and asymptomatic infection (E), respectively. Each predicted data point (blue), along with the median absolute risk and 95% credibility intervals (ribbon), are shown. The overall risk of a control group with no vaccination or prior infection history in this cohort (0.018) (green solid line) and the 90% reduction of relative risk (i.e., 10% relative risk [10%RR]) (green dotted line) are shown. (B, D, F) The combined effect of anti-N and anti-S antibody levels during the observation period on the absolute risk of both symptomatic and asymptomatic infection (B), symptomatic infection (D), and asymptomatic infection (F), respectively. The logarithmic absolute risk of infection

is indicated by the color bar. The white dotted and solid lines show the \log_{10} absolute risk decrease for every 0.5 and 1.0, respectively.

-The reliance on antibody levels as a sole measure of immunity can be limiting. T-cell responses and other aspects of the immune system are not considered, which could play significant roles in protection.

Reply to comment:

We fully agree with the reviewer, but our study aimed to explore serum anti-viral antibody levels correlated with protection against SARS-CoV-2 infections in hybrid immunity holders through a large-scale seroepidemiological survey. Due to the study design and costs, analysis of T-cell responses or other immunity, including antibody-dependent cellular cytotoxicity (ADCC) and antibody-dependent cellular phagocytosis (ADCP), was not possible. We added this as a limitation in the revised manuscript as follows:

Lines 336–337:

Fifth, we did not evaluate T-cell immunity and Fc-effector function in humoral immunity, which could play significant roles in protection against infection.

-The study uses a four-fold increase in anti-N antibody levels as a criteria for identifying re-infections, which could be arbitrary and might not account for individual variations in immune responses.

Reply to comment:

We apologize for any misunderstanding caused by our inadequate description. To avoid an arbitrary definition of serological re-infection in this study, we modeled anti-N antibody level dynamics to estimate the expected increase in anti-N antibody levels after re-infections within the two-month observation period (Figure 1C and 1D in the revised manuscript). To emphasize these results and avoid misinterpretation by the readers, we provided a detailed explanation of this analysis in the earlier part of the Result section of the revised manuscript (Lines 76–92, Figures 1C and 1D).

Lines 76–92:

We assessed the dynamics of anti-N antibody levels in individuals diagnosed with primary infection or re-infection to understand the differences in anti-N antibody responses and define the increase in anti-N antibody in SARS-CoV-2 re-infection. We applied a statistical model to assess the anti-N antibody response dynamics after diagnosis for all primary infection and re-infection cases using data obtained from cases with only the listed diagnosis date (primary infection, $n = 1218$; re-infection, $n = 26$) (Fig. 1C). For primary infection, anti-N titers were estimated to peak 69 days post-diagnosis and drop below the positive threshold (1.0 COI) at 621 days (95% credible interval (CredI), 522–754). For re-infection, the peak level of anti-N-antibody was estimated

as 4.8 times higher than that of primary infection, and the duration of antibody levels above 1.0 COI was estimated to be longer than that of primary infection. Using this model, we calculated the fold-increase in anti-N antibody levels from the pre-re-infection level to the level until 200 days post-re-infection (**Fig. 1D**). Since the interval from the initial to the second test in this cohort was approximately two months, the lowest fold-increase in anti-N antibody level from the pre-re-infection state to 60 days after re-infection was estimated to be four-fold or higher. These anti-N antibody response dynamics in those who were reinfected were consistent with previous reported data using the same anti-N antibody detection kit.^{30,31} These results suggest that a four-fold increase in serum anti-N antibody levels from the initial test to the second test in this cohort was considered a re-infection during the observation period.

However, as pointed out by the reviewer, the estimated anti-N antibody increase after re-infection in this study represented the overall trend in the study population and did not account for individual variations in anti-N antibody responses. While the criterion with a four-fold increase in anti-N antibody levels, which was determined by our anti-N antibody dynamics model, is expected to detect the most re-infection cases in the study population, there is a possibility of failure to detect re-infection cases whose anti-N antibody titer has not increased four-fold. We have added the description of this limitation in the revised manuscript as follows:

Lines 328–332:

Third, the estimated anti-N antibody increase after re-infection in this study represented the overall trend in the study population and did not account for individual variations in anti-N antibody responses. While the criterion with a four-fold increase in anti-N antibody levels, which our anti-N antibody dynamics model determined, is expected to detect the most re-infection cases in the study population, there is a possibility of failing to detect re-infection cases whose anti-N antibody titer has not increased four-fold.

Major comments:

-All samples from the cohort (non- and hybrid immunity individuals) were tested for anti-S and anti-N, but not for live or pseudovirus neutralization (NT50). Although anti-S and anti-N were expressed as a cutoff index, it would be crucial to report the IC50 titers from nAb, which would provide estimated values for 50% of protection. It would be nice to have a comparison of the antibody levels and nAb titers pre-reinfection for cases with control sample taken at a similar period of time. It would be nice to compare neutralization titers and their longitudinal dynamics in individuals with hybrid immunity or not.

Reply to comment:

We fully agree with the reviewer. Following the reviewer's suggestion, we subsampled the study participants and performed neutralization assays with the BA.5 virus to estimate infection risk

reduction by the neutralizing antibody titers (NT) against the BA.5 virus (Fig. 4 and 5). Subsampling strategies are described in Fig. 1B and the Methods section. The analysis utilizing BA.5 neutralizing titers exhibited a comparable pattern to that of S antibody titers, demonstrating that BA.5 neutralizing titers were associated with a reduced risk of infection. Detailed description has been added in Lines 191–213 and 221-226.

Lines 191-213:

Estimating infection risk reduction based on BA.5 neutralizing antibody titers and anti-N antibody levels

To estimate the risk of infection using neutralizing antibody titers (NT) against BA.5, we performed a stratified random sampling of 7–9% of newly infected and non-infected individuals during the observation period, both for those with and without hybrid immunity (N = 340, **Fig. 1B and Table S3**). Using the subsampled population, we estimated infection risk based on anti-S antibody titers and anti-N antibody levels. The estimated infection risks derived from the antibody titers in the subsampled population were consistent with those observed in the overall cohort for both symptomatic and asymptomatic infections, indicating that the subsampling had a low effect on estimating infection risk based on antibody levels (**Fig. 3A-F and Fig. S2**). Subsequently, infection risks were estimated using BA.5 NT and anti-N antibody levels in the subsampled population (**Fig. 4A–F**). The conditional effects of anti-N antibody levels indicated that the risk of infection decreased with increasing anti-N antibody levels for both symptomatic and asymptomatic infections, similar to the estimations based on anti-S antibody titers and anti-N antibody levels, shown in Fig. S2 (**Fig. 4A, C, E**). However, the conditional effects of BA.5 NT exhibited distinct patterns depending on symptom onset. For asymptomatic infections, the association between BA.5 NT and reduced infection risk was weak or negligible (**Fig. 4E and F**). In contrast, for symptomatic infections, the risk of infection clearly decreased with increasing BA.5 NT, and a substantial reduction in infection risk was observed when both anti-N antibody levels and BA.5 NT were high (**Fig. 4C, D**). For risk estimation of symptomatic infection only, the threshold for 90% protection was 10 BA.5 NT (**Table 4**). These findings indicate that serum anti-N antibody levels were strongly associated with reductions in the risk of both symptomatic and asymptomatic infections. However, anti-S antibody titers against the ancestral strain and neutralizing antibody titers against circulating virus were primarily associated with reducing only symptomatic infection risk.

Lines 221-226:

Similarly, the conditional effect of BA.5 NT or the combined effect of anti-N antibody levels did not show an association between high BA.5 NT and a reduced risk of re-infection (**Fig. 5C, D**). Given that anti-N antibody levels reflect the level of the immune response induced after viral infection,³² the level of serum anti-N antibodies determined using this method is a more reliable immunological correlate than the level of anti-S

antibodies and NTs in determining the effectiveness of preventing SARS-CoV-2 re-infection in individuals with hybrid immunity.

The study on the efficacy of COVID-19 vaccine during early pandemic by Gilbert et al. (Science 375, 43-50 (2022)) demonstrated a 90% protection antibody titer threshold, and the comparison of the estimation in our study with the previous reported threshold is expected to provide a more useful information. We summarized the corresponding antibody levels associated with 90% protection in Table 4, as follows:

Lines 208–209:

For risk estimation of symptomatic infection only, the threshold for 90% protection was 10 BA.5 NT (Table 4).

We compared the anti-S and anti-N antibody levels of newly infected hybrid immunity holders with those of non-infected hybrid immunity holders at the pre-reinfection initial test (December 2022 survey) with those of non-infected hybrid immunity holders (Fig. 2D and 2E), demonstrating that anti-N antibody levels of newly infected individuals at initial test were lower than those of non-infected individuals (Fig 2D), but anti-S antibody titer of newly infected individuals at initial test was comparable to those of non-infected individuals (Fig 2E). Following the reviewer's suggestion, we compared the neutralizing antibody titers against the ancestral and BA.5 viruses of newly infected hybrid immunity holders with those of non-infected hybrid immunity holders at the pre-reinfection initial test. This additional assay revealed that the neutralizing antibody titers against the ancestral and BA.5 viruses showed negligible differences between newly infected hybrid immunity holders and non-infected hybrid immunity holders, similar to the anti-S antibody titer (Fig. S1), and the information is stated as follows:

Lines 133–143:

Notably, for hybrid immunity holders, anti-N antibody levels of newly infected individuals at the initial test were lower than those of non-infected individuals (Fig 2D). The anti-S antibody titer of newly infected individuals at the initial test was comparable to those of non-infected individuals (Fig. 2E). For evaluating neutralizing antibody titers to the ancestral strain and BA.5 at the initial and second tests, we randomly selected 20 newly infected individuals with hybrid immunity and 20 without hybrid immunity. Furthermore, we performed propensity score matching for these 20 individuals to subsample 20 non-infected individuals during the observation period for each group (Fig.1B, Table S2). The neutralization assay with selected individuals revealed that the neutralizing antibody titers against the ancestral strain and BA.5 at the initial test showed indistinguishable differences between newly infected hybrid immunity holders and non-infected hybrid immunity holders, similar to the anti-S antibody titer (Fig. S1).

We compared the neutralizing antibody titers against the ancestral and BA.5 viruses between the initial and second tests in hybrid and non-hybrid immunity holders without infection during the observation period (Fig. 2F). The analysis revealed that neutralizing antibody titers against the ancestor strain and BA.5 exhibited minimal variation from the initial test to the second test in both hybrid immunity and non-hybrid immunity holders as follows:

Lines 150–152:

Neutralizing antibody titers against the ancestral strain and BA.5 exhibited minimal variation from the initial test to the second test in both hybrid immunity and non-hybrid immunity individuals who remained uninfected during the observation period (Fig. 2F).

-Authors do not analyze trajectories of antibody levels and neutralization titers before and after reinfection.

Reply to comment:

Following the reviewer's suggestion, we evaluated the antibody responses, including anti-S antibodies, anti-N antibodies, and neutralizing antibody titers against the ancestral strain and BA.5, before (initial test) and after (second test) infection in newly infected hybrid and non-hybrid immunity holders, including those with re-infections (Fig. 2F). Detailed description has been added as follows:

Lines 143–150:

Furthermore, in individuals without hybrid immunity, the geometric mean fold increase in neutralizing antibody titers against the ancestral strain and BA.5 from the initial to the second testing was 9.8- and 32.6-fold, respectively (Fig. 2F). In contrast, individuals with hybrid immunity demonstrated elevated neutralizing antibody titers against both the ancestral strain and BA.5 at the initial test (December 2022 survey) compared to those without hybrid immunity, and the increase in neutralizing antibody titers against the ancestral strain and BA.5 from the initial to the second tests for hybrid immunity holders was comparatively smaller (0.9-fold to the ancestral strain, and 2.7-fold to BA.5) than that observed in non-hybrid immunity holders.

-Authors should show the correlation between neutralization assays and antibody levels (Anti-S and anti-N).

Following the reviewer's suggestion, we evaluated the correlation between neutralizing antibody titers to the ancestral strain (Ancestral NT), neutralizing antibody titer to BA.5 (BA.5 NT), anti-N antibody levels, and anti-S antibody levels in hybrid and non-hybrid immunity holders (Fig. 2G). This analysis demonstrated that the anti-S antibody titer, the neutralizing antibody titer against the ancestral strain, and the neutralizing antibody titer against BA.5 were positively correlated with each other in both hybrid and non-hybrid immunity holders, while anti-N antibody levels were not correlated with any

other antibody titers, suggesting that anti-N antibody levels serve as a marker of antiviral immunity, independent of neutralizing antibody titers. Details have been added as follows:

Lines 152–158:

Next, the correlations between antibody titers in hybrid and non-hybrid immunity holders at each time point were evaluated (**Fig. 2G**). At any time point, the anti-S antibody titer, the neutralizing antibody titer against the ancestral strain, and the neutralizing antibody titer against BA.5 were positively correlated with each other in both hybrid and non-hybrid immunity holders, while anti-N antibody levels did not correlate with any other antibody titers, suggesting that anti-N antibody levels serve as a marker of antiviral immunity, independent of neutralizing antibody titers.

- Global SARS-CoV-2 seroprevalence has been observed to have substantial variation in the proportion of immunity induced by infection or vaccination in different settings worldwide (Bergeri I et. al. published). Authors should describe the seroprevalence for both hybrid immunity patients and non-hybrid immunity.

Reply to comment:

We agree with the reviewer. Following the reviewer's suggestion, we described the seroprevalence for both hybrid immunity and non-hybrid immunity holders in the Result section (Lines 68–75). The prevalence of anti-S antibodies in the five prefectures in Japan was previously reported at 97.2–98.3% in the December 2022 survey and 98.1–98.8% in the February to March 2023 survey. The prevalence of anti-N antibodies was reported at 17.6–28.8% in the December 2022 survey and 22.6–35.8% in the February 2023 survey. The prevalence of anti-S and anti-N antibodies in the enrolled participants are shown in Table 1 as follows:

Lines 68–75:

The prevalence of anti-S antibodies in the five prefectures in Japan was previously reported at 97.2–98.3% in the December 2022 survey and 98.1–98.8% in the February to March 2023 survey.^{26,27} The prevalence of anti-N antibodies was reported at 17.6–28.8% in the December 2022 survey and 22.6–35.8% in the February 2023 survey.^{26,27} Detailed seroprevalence has been reported in Japan in the 2021–2022 and 2023 surveys.^{28,29} The prevalence of anti-S and anti-N antibodies in the enrolled participants is shown in **Table 1**. The prevalence of anti-N antibodies in the hybrid immunity holder was $\geq 95\%$ in both surveys. The prevalence of anti-N antibodies in the non-hybrid immunity holder increased from 2.6% in the December 2022 survey to 11.8% in the February to March 2023 survey.

-The duration of the antibody response against SARS-CoV-2 differs broadly across individuals. Therefore, it is important to measure the peak of antibody titers at baseline and over time. It is possible that the intensity of the initial antibody response allows the estimation of long term antibody duration.

Reply to comment:

We thank the reviewer for their constructive comments and thoughtful suggestions. As the reviewer pointed out, the response of the anti-SARS-CoV-2 antibody differs broadly across individuals, and it is important to measure the peak of antibody titers at baseline and over time to estimate the anti-SARS-CoV-2 antibody duration and dynamics. Therefore, to properly predict the dynamics of individual anti-N antibody levels by statistical or mathematical modeling, it is essential to collect a minimum of three blood samples per case, including those with peak antibody titers. However, in our seroepidemiological study, serum collection was at two designated time points, December 2022 and February 2023, without considering the interval since vaccination or SARS-CoV-2 infections for each participant, as described in Fig. 1A. The peak antibody titers of nearly all individuals in our study were undetermined, complicating mathematical modeling. The anti-N antibody dynamics model shown in Fig. 1C was constructed using data from previously infected individuals who reported their COVID-19 diagnosis date, specifically primary infection cases (n=1,218) and reinfection cases (n=26). However, the exact timing of infection is unknown for the remaining previously infected and newly infected individuals, making it challenging to estimate the anti-N antibody response for each participant, as the reviewer suggested. The estimated anti-N antibody increase after re-infection in this study represented the overall trend in the study population and did not account for individual variations in anti-N antibody responses. While the criteria with a four-fold increase in anti-N antibody levels, which were determined by our anti-N antibody dynamics model, are expected to detect most re-infection cases in the study population, failure to detect re-infection cases whose anti-N antibody titer has not increased four-fold is possible. We have added this limitation in the revised manuscript as follows:

Lines 328-332:

Third, the estimated anti-N antibody increase after re-infection in this study represented the overall trend in the study population and did not account for individual variations in anti-N antibody responses. While the criterion with a four-fold increase in anti-N antibody levels, which our anti-N antibody dynamics model determined, is expected to detect the most re-infection cases in the study population, there is a possibility of failing to detect re-infection cases whose anti-N antibody titer has not increased four-fold.

The discussion should be more elaborate, including other studies pre-omicron or after Omicron (newly variants), in which anti-S showed protection.

Reply to comment:

Following the reviewer's suggestion, we added the description of additional studies that have d the protective role of anti-S antibodies or neutralizing antibodies before and after the emergence of Omicron variants in the introduction and discussion section (Lines 25–31, 293–296, 298–300, 304–312). These include research on vaccine efficacy from the pre-Omicron period and recent studies post-Omicron, which investigated neutralizing antibody or anti-S antibody titers correlates of protection against Omicron BA.1/2 (Sun et al. Nat Med. 2024; Sumner et al. J Infect Dis. 2024). Additionally, we checked PubMed for studies published between January 1, 2022, and August 20, 2024, using the search terms “SARS-CoV-2” in combination with the search terms “antibody,” “Omicron,” AND “Correlate(s) of Protection.” However, the immune correlates of protection against re-infection, especially among hybrid immunity holders with a history of infections and vaccination, remained unclear before our study, as described below.

Lines 25–31:

During the pre-Omicron epidemic period, anti-spike (S) antibody titers against the ancestral strain induced by vaccination were identified as immunological correlates for protection against SARS-CoV-2 infection and severe disease.^{4,9,10} While the prevention of severe disease through vaccination was confirmed even during the Omicron BA.1/2 and BA.4/5 epidemic periods,¹¹ a higher anti-spike (S) antibody titer against the ancestral strain was required for protection against infection during these periods compared to the pre-Omicron epidemic period.¹²⁻¹⁶

Lines 293–296:

Gilbert et al. reported that mRNA-1273 vaccine efficacy increased with anti-S IgG levels; for anti-S IgG titers of 33, 300, and 4000 BAU/ml, the vaccine efficacies were 85% (31 to 92%), 90% (77 to 94%), and 94% (91 to 96%), respectively.⁴

Lines 298–300:

Acute-phase ancestral anti-S antibody levels associated with 50% lower odds of COVID-19 were 1968 BAU/mL against Delta and 3375 BAU/mL against Omicron 1/2, showing that the required anti-S antibody levels for preventing symptomatic infection were increased for Omicron.¹⁶

Lines 304–312

Even in a study using neutralizing antibody titer for assessing the correlation with re-infection risk, the association between the anti-BA.1 neutralizing antibody titer in prior-infected individuals and protection against re-infection during Omicron BA.1/2 period was limited to 11%.⁴² Our results estimated 10209 BAU/mL of anti-

S antibody titer and 10 BA.5 NT had a 90% relative risk reduction against symptomatic infection. However, we did not find a 90% relative risk reduction for asymptomatic infection or infection in individuals with hybrid immunity, even at anti-S antibody titers above 100,000 BAU/mL and surpassing 3,000 BA.5 NT during the Omicron BA.5 epidemic period in this study. This may indicate that the protection conferred by existing COVID-19 vaccines is insufficient to prevent asymptomatic infection with the currently circulating omicron sub-lineages.

Reviewer #2 (Remarks to the Author):

Overall comments

1. A strength of the study is that it was done over a short period of time in the known Omicron BA.5 epidemic.

Reply to comment:

We would like to thank the reviewer for recognizing the value of our study. As you noted, research on correlates of protection during and after the Omicron BA.5 epidemic is scarce, and we appreciate your positive evaluation.

2. A slight weakness is only measuring antibodies to the Ancestral strain instead of also to Omicron BA.5 that would enable studying variant-matched correlates. I believe it is only a slight weakness given literature supporting that Ancestral strain antibodies tend to be as good as correlate as variant-matched antibodies.

Reply to comment:

We would like to thank the reviewer for the thoughtful evaluation of our study and fully agree with the comments. Following the other reviewer's suggestion, we subsampled the study participants and performed neutralization assays with BA.5 virus on the subsampled participants to estimate infection risk reduction by the neutralizing antibody titers (NT) against BA.5 virus (Fig. 4 and 5). Subsampling strategies are described in Fig. 1B and the Methods section. The analysis utilizing BA.5 neutralizing titers exhibited a comparable pattern to that of S antibody titers, demonstrating that BA.5 neutralizing titers were associated with a reduced risk of infection. Details description have been added as follows:
Lines 191–213:

Estimating infection risk reduction based on BA.5 neutralizing antibody titers and anti-N antibody levels

To estimate the risk of infection using neutralizing antibody titers (NT) against BA.5, we performed a stratified random sampling of 7–9% of newly infected and non-infected individuals during the observation period, both for those with and without hybrid immunity (N = 340, Fig. 1B and Table S3). Using the subsampled population,

we estimated infection risk based on anti-S antibody titers and anti-N antibody levels. The estimated infection risks derived from the antibody titers in the subsampled population were consistent with those observed in the overall cohort for both symptomatic and asymptomatic infections, indicating that the subsampling had a low effect on estimating infection risk based on antibody levels (**Fig. 3A-F and Fig. S2**). Subsequently, infection risks were estimated using BA.5 NT and anti-N antibody levels in the subsampled population (**Fig. 4A-F**). The conditional effects of anti-N antibody levels indicated that the risk of infection decreased with increasing anti-N antibody levels for both symptomatic and asymptomatic infections, similar to the estimations based on anti-S antibody titers and anti-N antibody levels, shown in Fig. S2 (**Fig. 4A, C, E**). However, the conditional effects of BA.5 NT exhibited distinct patterns depending on symptom onset. For asymptomatic infections, the association between BA.5 NT and reduced infection risk was weak or negligible (**Fig. 4E and F**). In contrast, for symptomatic infections, the risk of infection clearly decreased with increasing BA.5 NT, and a substantial reduction in infection risk was observed when both anti-N antibody levels and BA.5 NT were high (**Fig. 4C, D**). For risk estimation of symptomatic infection only, the threshold for 90% protection was 10 BA.5 NT (**Table 4**). These findings indicate that serum anti-N antibody levels were strongly associated with reductions in the risk of both symptomatic and asymptomatic infections. However, anti-S antibody titers against the ancestral strain and neutralizing antibody titers against circulating virus were primarily associated with reducing only symptomatic infection risk.

Lines 221–226:

Similarly, the conditional effect of BA.5 NT or the combined effect of anti-N antibody levels did not show an association between high BA.5 NT and a reduced risk of re-infection (**Fig. 5C, D**). Given that anti-N antibody levels reflect the level of the immune response induced after viral infection,³² the level of serum anti-N antibodies determined using this method is a more reliable immunological correlate than the level of anti-S antibodies and NTs in determining the effectiveness of preventing SARS-CoV-2 re-infection in individuals with hybrid immunity.

3. Multiple infectious disease outcomes are studied such as new symptomatic COVID-19, new asymptomatic infection, and new infection regardless of symptoms. I had a hard time finding the numbers of endpoints/outcomes of the different types. It may strengthen the article to add more dissection/description of the numbers of different outcomes (stratified by key covariate groups). This issue surfaced especially at page 11, lines 305-306.

Reply to comment:

We fully agree with the reviewer. Following the reviewer's suggestion, we have added separate analyses for symptomatic infection, asymptomatic infection, and infections, including symptomatic

and asymptomatic cases in Fig. 3 and Fig. 4 (Lines 164–180, 191–213). The corresponding numbers of participants for each outcome have been added to Table S1. We are extremely grateful for this insightful and constructive comment. We believe that this additional analysis has greatly increased the robustness and value of our study.

Lines 164–180:

Conditional effects of the anti-N antibody levels indicated that the risk of infection decreased logarithmically with increasing anti-N antibody levels in infection regardless of symptoms and symptomatic and asymptomatic infections (Fig. 3A, C, E). In symptomatic and asymptomatic infections, individuals with more than 1.0 COI and 17.2 anti-N antibody levels had a 90% relative risk reduction of new infections compared with a control group with no vaccination or prior infection history, respectively (Table 4). As shown above, anti-N antibody levels peaked 1–2 months post-infection and declined over time (Fig. 1C). The estimated median duration for which anti-N antibody levels remained over 17.2 COI preventing for asymptomatic infection was 117 (CredI, 99–132) days after primary infection and 505 (CredI, 276–1000) days after re-infection, which is markedly shorter than the duration in the threshold 1.0 COI preventing for symptomatic infection as described above (Fig. 1C). While higher anti-S antibody titers were associated with a decreased risk of infection, the impact of this relationship was modest (Fig. 3A, C, E). In the risk estimation for symptomatic infection only, the threshold for 90% protection was estimated at 10,209 BAU/mL. Even at the highest observed anti-S antibody titer of 523,000 BAU/mL, the estimated reduction in relative risk was only 70% for asymptomatic infection. Evaluation of the combined impact of anti-N and anti-S antibody levels revealed a modest decrease in the absolute risk of infection attributable to anti-S antibody titers, whereas a reduction in the absolute risk of infection was significantly associated with an increase in anti-N antibody levels (Fig. 3B, D, F).

Lines 191–213:

Estimating infection risk reduction based on BA.5 neutralizing antibody titers and anti-N antibody levels

To estimate the risk of infection using neutralizing antibody titers (NT) against BA.5, we performed a stratified random sampling of 7–9% of newly infected and non-infected individuals during the observation period, both for those with and without hybrid immunity (N = 340, Fig. 1B and Table S3). Using the subsampled population, we estimated infection risk based on anti-S antibody titers and anti-N antibody levels. The estimated infection risks derived from the antibody titers in the subsampled population were consistent with those observed in the overall cohort for both symptomatic and asymptomatic infections, indicating that the subsampling had a low effect on estimating infection risk based on antibody levels (Fig. 3A-F and Fig. S2). Subsequently, infection risks were estimated using BA.5 NT and anti-N antibody levels in the subsampled population (Fig. 4A–F). The conditional effects of anti-N antibody levels indicated that the risk of infection decreased with increasing anti-N antibody levels for both symptomatic and asymptomatic infections, similar to the estimations based on anti-S antibody titers and anti-N antibody levels, shown in Fig. S2 (Fig. 4A, C, E). However, the conditional effects

of BA.5 NT exhibited distinct patterns depending on symptom onset. For asymptomatic infections, the association between BA.5 NT and reduced infection risk was weak or negligible (**Fig. 4E and F**). In contrast, for symptomatic infections, the risk of infection clearly decreased with increasing BA.5 NT, and a substantial reduction in infection risk was observed when both anti-N antibody levels and BA.5 NT were high (**Fig. 4C, D**). For risk estimation of symptomatic infection only, the threshold for 90% protection was 10 BA.5 NT (**Table 4**). These findings indicate that serum anti-N antibody levels were strongly associated with reductions in the risk of both symptomatic and asymptomatic infections. However, anti-S antibody titers against the ancestral strain and neutralizing antibody titers against circulating virus were primarily associated with reducing only symptomatic infection risk.

4. The manuscript supports the importance of integrating anti-N concentration data in immune correlates analyses of COVID-19 vaccines in populations that are largely previously infected. Could the reviewers put this result in the context of flu vaccine research (alternatively for other relevant pathogens) where immune correlates are also studied in previously infected populations? It would be valuable to communicate whether the main messages of this article regarding the improvement of immune correlates through including anti-core protein antibody responses (non-vaccine antigens) in addition to surface Envelope vaccine protein antibody responses carries over to other pathogens, and whether there is a history of this being effectively done.

Reply to comment:

We thank the reviewer for their constructive comments and thoughtful suggestions. The correlation between anti-nucleoprotein antibodies and a reduced risk of viral infection has not been reported in cases of influenza or other respiratory viruses, emphasizing the novelty of our findings. In human influenza research, the importance of nucleoprotein-specific CD8 and CD4 T cells in protection against influenza has been demonstrated (Sridhar et al., Nat Med. 2013; Wilkinson et al., Nat Med. 2012). Furthermore, studies on universal influenza vaccines targeting nucleoproteins have shown promising results, with Phase II clinical trials of the OVX836 vaccine demonstrating suppression of symptomatic influenza infections through NP-targeted immunity (Leroux-Roels et al., Lancet Infect Dis. 2023). For SARS-CoV-2, vaccines combining S and N proteins enhance immunogenicity and efficacy through CD8 T-cell-dependent mechanisms (Hajink et al. Sci Transl Med. 2022). We added the descriptions of these previous reports to provide a broader context for our findings on the roles of anti-nucleoprotein antibodies for infection prevention in the discussion section as follows:

Lines 264–273:

The correlation between anti-nucleoprotein antibodies and a reduced risk of viral infection has not been reported in cases of influenza or other respiratory viruses, emphasizing the novelty of our findings. In human influenza

research, the importance of nucleoprotein-specific CD8⁺ and CD4⁺ T cells in protection against influenza virus infection has been emphasized.^{36,37} Furthermore, studies on universal influenza vaccines targeting nucleoproteins OVX836 have shown suppression of symptomatic influenza infections through nucleoprotein-targeted immunity in Phase II clinical trials.³⁸ For SARS-CoV-2, vaccines combining S and N proteins can enhance immunogenicity and efficacy through CD8⁺ T-cell-dependent mechanisms in non-human primates.³⁹ Considering these studies alongside our findings, it is plausible that immunity targeting nucleoprotein/nucleocapsid is strongly associated with controlling respiratory viral infections.

5. The manuscript often reports on 'infection prevention efficacy' which is presumably efficacy from hybrid immunity (vaccination and prior infection) vs. vaccine immunity only (without prior infection), but I found the descriptions confusing at times. I'm not sure what 'infection prevention efficacy' refers to, where it could refer to the protection from a recent infection regardless of vaccination pattern. In general protection depends on both previous infection history and vaccination history, and I didn't think the way in which both of these factors were accounted for was sufficiently clearly described. Another way to say this comment, is I'm not sure that vaccination history information was accounted for as completely as may be helpful.

Reply to comment:

We deeply apologize for the confusion in our terminology. We used the term 'infection prevention efficacy' in the manuscript as "efficacy of preventing infection" associated with specific antibody titers, independent of vaccination history or prior infection status. As this reviewer noted, the primary objective of this study was to evaluate whether baseline antibody titers are associated with reduced infection risk, and we did not compare between hybrid immunity holders and non-hybrid immunity holders. We agree with the reviewer that the term 'infection prevention effect' may confuse readers because it may imply a comparison between hybrid and non-hybrid immunity. We apologize for the ambiguity and have revised the terminology throughout the manuscript to clarify this point.

6. Tied to the last comment, adjustment for vaccination status and vaccine type is relevant for confounding control and effect modification assessment; can more details be added for how the vaccination history variables are studied?

Reply to comment:

Vaccination history was considered using two variables: the number of vaccine doses received (categorized as 0 to 5 doses) and whether the individual had received an Omicron-adapted bivalent vaccine. In the infection risk estimation analysis using antibody titers, factors including vaccination history were incorporated as weights to account for potential sampling bias between newly infected

and non-infected individuals. This approach ensures a balanced analysis and minimizes bias related to vaccination history. We added the detailed description in the Methods section as follows:

Lines 483–485:

The predictors were age group (20–64 years or 65 years or older), biological sex, presence of comorbidities, Omicron BA.1/BA.5 bivalent vaccination, vaccination count (categorized as 0 to 5 doses), prior infection history, and municipality.

7. In the Statistical analysis section on page 12, use of logistic regression is described. However, elsewhere the manuscript describes that a Bayesian GAM and Poisson regression are also applied. The coordinated use of these 3 approaches is not described, ideally there would be a mapping of which method is applied to which objective and why, so the reader could easily see the rationale for this set of methods for meeting all of the objectives. It looks like this ‘Statistical analysis’ heading is meant to fit under ‘Modeling the antibody response’, but the way it is written it comes across as if it is the consolidated section describing statistical analyses.

Reply to comment:

We deeply apologize for the confusion in the description of statistical analysis. We revised the Methods section to clarify which analyses correspond to each figure, making the connection between the statistical methods and their objectives clearer (Lines 475–477, 480–492, 524–525).

Logistic regression was used in the original manuscript for infection risk estimation without antibody titers in Fig. 3E. However, in the revised manuscript, with the inclusion of infection risk estimation using BA.5 neutralizing antibody titers, we found logistic regression was no longer necessary for the primary discussion and have therefore removed it.

Lines 475–477:

Combined models were fitted for combinations of anti-N antibody levels with anti-S antibody titers (Fig. 3A–3F, 5A, 5B, and S1A–S1F) or BA.5 NT (Fig. 4A–4F, 5C, and 5B), controlling for baseline exposure risk, and weighted using inverse probability weights as described below.

Lines 480–492:

In addition to antibody titers, we assumed that each participant's absolute risk of new infections depends on the region and basic demographics. We used a Poisson regression model to predict the probability of infection, adjusting for the participants' baseline characteristics, and reduce potential bias in the availability of samples between newly infected and non-infected individuals. The predictors were age group (20–64 years or 65 years or older), biological sex, presence of comorbidities, Omicron BA.1/BA.5 bivalent vaccination, vaccination count (categorized as 0 to 5 doses), prior infection history, and municipality. We used the estimated probability as these inverse probability weights in the above GAM to adjust the potential bias from participants' baseline

characteristics to better isolate the association of immune correlates (antibody titers) on infection risk reduction. The infection risk estimation in the GAM was constructed using *brms* 2.21. In the GAM, a Poisson distribution was used for the response variable. The newly infected response variable was modeled as a function of smooth terms for combinations of anti-N antibody levels with anti-S antibody titers or BA.5 NT using cubic splines with three basis functions ($k = 3$).

Lines 524–525:

In the correlation matrix analyses, Spearman correlations between variables were calculated with false discovery rate (FDR) corrections (**Fig. 2G**), as described previously.³²

Specific comments:

1. Page 3, lines 52-54, “Additionally, recent reports...” Clarify this finding?

Reply to comment:

We apologize for the ambiguous description in the part you pointed out. We have revised the text as follows:

Lines 36–38:

Recent reports have shown that nasal mucosal secretory anti-spike IgA antibody levels, elicited after infection, are associated with prevention of the SARS-CoV-2 infection and infectious viral shedding in the upper respiratory tract.

2. Page 3, lines 69-70, I think what is meant is more like “association of ... antibody levels with the occurrence of a new infection.”

Reply to comment:

We apologize for the ambiguous description in the part you pointed out. We revised the text as follows:

Lines 51–54:

We evaluated the infection risk reduction associated with anti-S antibody titers and BA.5 neutralizing antibody titers and the combined effect of infection-related anti-N antibody levels during the Omicron BA.5 epidemic period by analyzing the association of combined serum antibody levels with the occurrence of a new infection.

3. Page 4, The data contribute information about the value of anti-N concentration for detecting new infections. Can the authors comment on how well anti-N concentration can be used to estimate the timing of infection?

Reply to comment:

We thank the reviewer for their constructive comments and thoughtful suggestions. To properly predict the dynamics of individual anti-N antibody levels by statistical or mathematical modeling, it is essential to collect a minimum of three blood samples per case, including those with peak antibody titers. However, in our seroepidemiological study, serum collection was at two designated time points, December 2022 and February 2023, without considering the interval since vaccination or SARS-CoV-2 infections for each participant, as described in Fig. 1A. The peak antibody titers of nearly all individuals in our study were undetermined, complicating mathematical modeling. The anti-N antibody dynamics model shown in Fig. 1C was constructed using data from previously infected individuals who reported their COVID-19 diagnosis date, specifically primary infection cases (n=1,218) and reinfection cases (n=26). However, the exact timing of infection is unknown for the remaining previously infected and newly infected individuals, making it challenging to estimate the anti-N antibody response for each participant, as the reviewer suggested. The estimated anti-N antibody increase after re-infection in this study represented the overall trend in the study population and did not account for individual variations in anti-N antibody responses. While the criterion with a four-fold increase in anti-N antibody levels, which was determined by our anti-N antibody dynamics model, is expected to detect the most re-infection cases in the study population, there is a possibility of failure to detect re-infection cases whose anti-N antibody titer has not increased four-fold. We have added the description of this limitation in the revised manuscript as follows:

Lines 328–332:

Third, the estimated anti-N antibody increase after re-infection in this study represented the overall trend in the study population and did not account for individual variations in anti-N antibody responses. While the criterion with a four-fold increase in anti-N antibody levels, which our anti-N antibody dynamics model determined, is expected to detect the most re-infection cases in the study population, there is a possibility of failing to detect re-infection cases whose anti-N antibody titer has not increased four-fold.

4. Page 5, lines 134-135: 'infection prevention efficacy' seems like a vague term, where just after this section heading it is explained that the efficacy of hybrid immunity vs. not hybrid immunity is clearer. Although this phraseology is not completely clear either, because it is isn't clear whether hybrid immunity is being compared to vaccinated not previously infected or to not vaccinated and not previously infected, or to a mixture (not previously infected regardless of vaccination history).

Reply to comment:

We deeply apologize for the confusing terminology. As mentioned above, we had used the term 'infection prevention efficacy' in the manuscript as "efficacy of preventing infection" associated with

specific antibody titers, independent of vaccination history or prior infection status. We apologize for the ambiguity and have revised the terminology throughout the manuscript to clarify this point. The relative risk shown in Fig. 3A and Fig. 4A represents a comparison to a 'No exposure' group, defined as individuals without prior infection or vaccination history. Therefore, the term "reduced risk of infection" refers to the decrease in infection risk that occurs in the hybrid or non-hybrid immunity groups compared to the no-exposure group. We added the description in the Methods section as follows:

Lines 500–502:

In this cohort, the overall risk of a control group with no vaccination or prior infection history included in the non-hybrid immunity holders, referred to as no exposures, was calculated using 16 newly infected individuals among 87 no exposures (0.184).

5. I think the first paragraph of the Discussion would be improved if it includes information on the timing of sample draws relative to outcomes, otherwise it is too vague of a summary.

Reply to comment:

Following the reviewer's suggestion, we revised the first paragraph of the Discussion to include information on the timing of sample draws relative to the outcomes, along with the findings obtained from the updated analysis, as follows:

Lines 229–243

In this study, we evaluated the protective effect of serum anti-N, anti-S, and Omicron BA.5 neutralizing antibody levels against SARS-CoV-2 infection during the Omicron BA.5 epidemic period from December 2022 to February 2023 in Japan. We showed that reducing symptomatic and asymptomatic infection risk was correlated with anti-N antibody levels at the initial test (December 2022), which were induced by previous infection before December 2022, on a logarithmic scale, even in hybrid immunity holders. In contrast, anti-S antibody titers against the ancestral spike antigen and BA.5 neutralizing antibody titers at the initial test (December 2022), which were induced by both vaccines and infections before December 2022, showed protective associations against only symptomatic infection during the BA.5 epidemic period. Interestingly, anti-S antibody titers against the ancestral spike antigen and BA.5 neutralizing antibody titers at the initial test exhibited no protective correlation against infection in individuals with hybrid immunity or against asymptomatic infection during the BA.5 epidemic period. These results suggest that the risk of re-infection can be assessed based on anti-N antibody titers in individuals with hybrid immunity. Moreover, this finding implies that infection-induced immunity in individuals with hybrid immunity includes a distinct and potent immunity beyond the presence of serum anti-S antibodies and neutralizing antibodies against the circulating strains, which correlates with serum anti-N antibody levels.

6. Page 7, line 186, I think 'non-mechanistic' is meant, not 'non-mechanical' ('non-mechanistic' is the term in the literature).

Reply to comment:

We sincerely apologize for the inadequate description. We corrected the term to 'non-mechanistic' according to the reviewer's suggestion.

7. Page 11, lines 305-306, and elsewhere in Methods. Aspects of the writing such as 'turn positive' is too imprecise, please explicitly write that the diagnosis is based on such and such data from information available at the second visit. Relatedly, in the Infection risk estimation section, lines 319-324, the description omits the information on time points at which antibody markers are measured, and/or case status is ascertained. As this kind of information is basic to the analysis, it should be directly described explicitly, rather than the way it is done now leaving some details tacit that the reader needs to infer. E.g., when it says the Poisson model is used to predict the probability of infection, it would be useful to state diagnosis of infection over what temporal period and diagnosed at what time point, etc.

Reply to comment:

Following the reviewer's suggestion, we added a schematic diagram of the study design, now Fig. 1A, to clarify that serum samples and self-administered questionnaires were collected both during the December 2022 and February 2023 surveys. Furthermore, we revised the descriptions in the Methods section to explicitly define newly infected individuals, newly symptomatic infected individuals, and newly asymptomatic individuals with precise details on the timing of sample collection and case status determination as follows:

Lines 388–406:

Participants with a history of self-test confirmed SARS-CoV-2 infection during the observation period, those who were diagnosed with COVID-19 during the observation period, or those with a seroconverted anti-N antibody from the initial to the second test (positive threshold, 1.0 COI) were considered newly infected cases. Participants who showed a four-fold or higher increase in anti-N antibody levels at the second antibody test compared to those at the initial test were considered newly infected, including those with re-infections (**Fig. 1B and 1D**). Specifically, participants with no previous COVID-19 diagnosis, or previous self-test-confirmed SARS-CoV-2 infection in the self-administered questionnaire at the time of the December 2022 survey (from November 26, 2022, to December 27, 2022) and whose serum anti-N antibody level at the December 2022 survey was below 1.0 COI, were considered uninfected participants at the beginning of the observation period.

Among the uninfected participants at the beginning of the observation period, those who subsequently reported a COVID-19 diagnosis or self-test confirmed SARS-CoV-2 infection at the February 2023 survey (from February 3, 2023, to March 4, 2023) or whose serum anti-N antibody level at the February 2023 survey was above 1.0 COI, were defined as newly infected cases during the observation period from December 2022 to February 2023. Among the prior-infected participants at the beginning of the observation period, those who reported a COVID-19 diagnosis, or self-test confirmed SARS-CoV-2 infection between the initial and the second survey were considered newly re-infected during the observation period. Based on the anti-N antibody fold-increase model for re-infections, participants whose serum anti-N antibody levels increased by four-fold or more from the December 2022 survey to the February 2023 survey were considered as newly re-infected.

Lines 413-425

Participants who met the criteria for newly infected individuals described above, along with COVID-19 symptoms described in the February 2023 survey questionnaires, were classified as newly symptomatic infected individuals. Participants with a history of self-test-confirmed SARS-CoV-2 infection or COVID-19 diagnosis during the observation period without COVID-19 symptoms described in the February 2023 survey questionnaires were classified as newly “asymptomatic SARS-CoV-2 test-confirmed infected” individuals, as shown in **Table 3**. Among asymptomatic infected individuals, participants without a history of COVID-19 diagnosis or self-test confirmed SARS-CoV-2 infection during the observation period but met the criteria for newly infected individuals through anti-N antibody level increases (seroconverted anti-N antibody (positive threshold, 1.0 COI) or a four-fold increase from the December 2022 survey to the February 2023 survey) were classified as newly “asymptomatic serological defined” infected individuals, as shown in **Table 3**. For infection risk estimation (**Fig. 3, 4, 5, S1, Table S1**), the asymptomatic infection category included both “asymptomatic SARS-CoV-2 test-confirmed” and “asymptomatic serological defined” infected individuals.

8. Page 11, lines 329-330. It is not clear how weighing the correlates of risk models eliminates bias, this needs more details.

Reply to comment:

Following the reviewer’s suggestion, we revised the manuscript to include more details on this aspect as follows:

Lines 480–488:

In addition to antibody titers, we assumed that each participant's absolute risk of new infections depends on the region and basic demographics. We used a Poisson regression model to predict the probability of infection, adjusting for the participants' baseline characteristics, and reduce potential bias in the availability of samples between newly infected and non-infected individuals. The predictors were age group (20–64 years or 65 years or older), biological sex, presence of comorbidities, Omicron BA.1/BA.5 bivalent vaccination, vaccination

count (categorized as 0 to 5 doses), prior infection history, and municipality. We used the estimated probability as these inverse probability weights in the above GAM to adjust the potential bias from participants' baseline characteristics to better isolate the association of immune correlates (antibody titers) on infection risk reduction.

9. Fig 3. Caption, "Estimation of the effect..." Consider changing the word 'effect' to 'association' given that antibody levels were not randomized and formal causal inference methodology was not employed that would enable pursuing the assessment of effects.

Reply to comment:

Following reviewer's suggestion, we changed the word "effect" to "association" to reflect the focus of this study on estimating associations rather than causality.

10. Fig. 3 shows very narrow confidence intervals, even in the right-tail of anti-N concentration where there is little data support. Therefore, this analysis seems to report more precision than is actually available. I think this is because of the parametric models that are used and precision is quantified assuming the parametric shapes are correct. Consider adding to Limitations in the discussion.

Reply to comment:

We thank the reviewer for their constructive comments. Although the GAM used in this study is a non-parametric model that estimates antibody titers using cubic splines, we restricted the number of basis functions ($k = 3$) to allow estimation in regions with sparse data. However, as you noted, the reliability of estimates in areas with sparse data may be low. Following the reviewer's suggestion, we added the description below to the limitation section.

Lines 333–335:

Fourth, for estimating the infection risk reduction based on antibody levels, the reliability of estimates in regions with sparse antibody titer data may be low. While cubic splines with limited basis functions ($k = 3$) were used to stabilize these estimates, this limitation should be considered when interpreting the results.

Point-by-Point Response to Reviewers' comments:

Thank you for providing invaluable feedback on our research. We have addressed each comment in a point-wise manner, indicated in blue text. Changes and additions to the main text have been highlighted in yellow for your ease of review.

Reviewer #1 (Remarks to the Author):

I thank the authors for addressing most of my concerns.

While I understand that the role of T-cell responses and mucosal immunity is beyond the immediate scope of the paper, elaborating on their contribution to hybrid immunity could enhance the discussion. This addition would align the findings more closely with the broader immunological context and provide further insights into the observed correlations.

Specifically:

1. Discussing the interplay between T-cell responses and anti-N antibody levels could offer a more comprehensive understanding of the mechanisms behind hybrid immunity.
2. Addressing mucosal immunity could strengthen the paper's relevance, particularly given its established role in asymptomatic and upper respiratory tract infections.

Reply to comment:

We would like to thank the reviewer for the opportunity to further evaluate and discuss the immunological aspects of our study. In particular, regarding the two points raised in your comments, we have provided the following responses and have incorporated additional discussion in the Discussion section.

1. Discussing the interplay between T-cell responses and anti-N antibody levels could offer a more comprehensive understanding of the mechanisms behind hybrid immunity.

We would like to thank the reviewer for their helpful comment. In COVID-19, several reports have indicated partial interplays between T cell responses and anti-N antibody levels. The non-hospitalized patients had higher CXCR5⁺ CD4⁺ T follicular helper cell responses than hospitalized ones, and virus-specific CD4⁺ T cell responses correlated positively with serum anti-N and anti-S antibody levels within 180 days post-onset (Nelson et al., *Sci Immunol*, 2022). In breakthrough infections after vaccination, which the majority of individuals with hybrid immunity, an increased CD4⁺ T cell responses to various SARS-CoV-2 proteins including N proteins has been reported (Tarke et al., *Cell Rep Med*, 2024), suggesting that the enhanced CD4⁺ T cell response against non-spike proteins may be reflected in higher serum anti-N antibody levels in individuals with hybrid immunity. Moreover, SARS-CoV-2 N proteins are expressed on the cell surface of infected cells, making it a potential target for anti-N antibodies mediated destruction of infected cells

(López-Muñoz et al., Sci Adv, 2022). Similar phenomena have been reported with other RNA viruses, including Influenza virus (Yewdell et al., J Immunol, 1981), where anti-nucleoprotein antibodies mediate lysis of virus infected cells (Staerz et al., Eur J Immunol, 1987) and provide infection protection in mice models (Fujimoto et al., J Gen Virol, 2016; LaMere et al., J Immunol, 2011). For SARS-CoV-2 infection, *in vitro* studies have shown that cell surface N protein expression can induce antibody-dependent cellular cytotoxicity (ADCC) via the anti-N antibody Fc domain (López-Muñoz et al., Sci Adv, 2022). In addition, treatment with anti-N sera or monoclonal antibodies suppressed viral replication by inducing NK cell-mediated ADCC in mice models (Dangi et al., J Clin Invest, 2022). Therefore, N is a promising target vaccine antigen to induce immunity resistant to SARS-CoV-2 immune evasions. Following the reviewer's suggestion, we added these descriptions to the Discussion section in the revised manuscript as follows:

Lines 475–471:

Regarding T cell immunity, the non-hospitalized COVID-19 cases had higher CXCR5⁺ CD4⁺ T follicular helper cell responses than hospitalized cases, and virus-specific CD4⁺ T cell responses correlated positively with serum anti-N and anti-S antibody levels within 180 days post-onset.³⁹ In breakthrough infections after vaccination, which the majority of individuals with hybrid immunity, an increased CD4⁺ T cell responses to various SARS-CoV-2 proteins including N proteins has been reported,⁴⁰ suggesting that the enhanced CD4⁺ T cell response against non-spike proteins may be reflected in higher serum anti-N antibody levels in individuals with hybrid immunity.

Lines 481–488:

Moreover, SARS-CoV-2 N proteins are expressed on the cell surface of infected cells, making it a potential target for anti-N antibodies mediated destruction of infected cells.⁴⁵ Similar phenomena have been reported with other RNA viruses, including Influenza virus,⁴⁶ where anti-nucleoprotein antibodies mediate lysis of virus infected cells⁴⁷ and provide infection protection in mice models.^{48,49} For SARS-CoV-2 infection, *in vitro* studies have shown that cell surface N protein expression can induce antibody-dependent cellular cytotoxicity (ADCC) via the anti-N antibody Fc domain.⁴⁵ In addition, treatment with anti-N sera or monoclonal antibodies suppressed viral replication by inducing NK cell-mediated ADCC in mice models.⁵⁰

2. Addressing mucosal immunity could strengthen the paper's relevance, particularly given its established role in asymptomatic and upper respiratory tract infections.

Our results showed that asymptomatic infections occurred more proportionally in newly infected individuals with hybrid immunity than those without hybrid immunity, suggesting that individuals with hybrid immunity possess robust protective mechanisms that effectively prevent the symptom onset. On the upper respiratory tract, anti-viral mucosal secretory IgA antibodies are produced within few days after onset (Miyamoto et al., PNAS, 2023). This mucosal antibody response is thought to restrict the expansion of viral infection to the lower respiratory tract, potentially resulting

in asymptomatic or mild infections (Russell et al., Front Immunol, 2022). It was also reported that nasal anti-S IgA responses after onset were correlated with lower viral loads and symptom resolution (Fröberg et al., Nat Commun, 2021). Our previous work also showed that reinfected individuals with prior infection mounted a faster and higher nasal secretory IgA response, which was associated with shorter periods of viral shedding.²² However, our previous study showed that no correlation was observed between the early secretory IgA response and the symptom onset or duration of symptomatic period; thus, it remains unclear whether the mucosal antibody response directly links to the symptom prevention (Miyamoto et al., PNAS, 2023). The current study did not collect any mucosal specimen, and the relationship between infection-induced mucosal immunity and symptom prevention remains unclear. Further research is needed to elucidate immunological factors involved in symptom prevention and their connection to mucosal immunity in individuals with hybrid immunity. Following the reviewer's suggestion, we added these descriptions to the Discussion section in the revised manuscript as follows:

Lines 491-505:

Our results showed that asymptomatic infections occurred more proportionally in newly infected individuals with hybrid immunity than those without hybrid immunity, suggesting that individuals with hybrid immunity possess robust protective mechanisms that effectively prevent the symptom onset. On the upper respiratory tract, anti-viral mucosal secretory IgA antibodies are produced within few days after onset.²² This mucosal antibody response is thought to restrict the expansion of viral infection to the lower respiratory tract, potentially resulting in asymptomatic or mild infections.⁵¹ It was also reported that nasal anti-S IgA responses after onset were correlated with lower viral loads and symptom resolution.⁵² Our previous work also showed that reinfected individuals with prior infection mounted a faster and higher nasal secretory IgA response, which was associated with shorter periods of viral shedding.²² However, our previous study showed that no correlation was observed between the early secretory IgA response and the symptom onset or duration of symptomatic period; thus, it remains unclear whether the mucosal antibody response directly links to the symptom prevention.²² The current study did not collect any mucosal specimen, and the relationship between infection-induced mucosal immunity and symptom prevention remains unclear. Further research is needed to elucidate immunological factors involved in symptom prevention and their connection to mucosal immunity in individuals with hybrid immunity.